# Cross ionization mode chemical similarity prediction between tandem mass spectra in metabolomics

Niek F. de Jonge [1] ✉, Elena Chekmeneva [2], Robin Schmid [3,4], David Joas [5], Lem-Joe Truong[5], Justin J. J. van der Hooft [1,6,7] ✉ & Florian Huber [5,7] ✉

Mass spectrometry is a cornerstone of untargeted metabolomics, enabling the characterization of metabolites in both positive and negative ionization modes. However, comparisons across ionization modes have remained a substantial challenge due to the distinct fragmentation patterns produced by each polarity. To overcome this barrier, we present MS2DeepScore 2.0, a machine learning-based model to predict chemical similarity between mass fragmentation spectra, which works both between different and the same ionization modes. We demonstrate the utility of MS2DeepScore 2.0 in three case studies, where MS2DeepScore enabled cross-ionization mode molecular networking, enhancing data exploration and metabolite annotation. To ensure robustness, we have implemented a quality estimation method that flags spectra with low information content or those dissimilar to the training data, thereby minimizing false predictions. Altogether, MS2DeepScore 2.0 extends our current capabilities in organizing, exploring, and annotating untargeted metabolomics profiles.

Mass spectrometry is widely used to map the chemical contents of natural extracts and other biological mixtures. In untargeted metabolomics, tandem mass spectrometry (or mass spectrometry fragmentation, MS/MS, MS[2]) is typically used to support structural annotation of metabolite features detected in metabolomics profiles. Interpretation of tandem mass spectra is increasingly done with the help of computational tools that assist with structurally annotating mass spectra, such as SIRIUS[1], MS-Finder[2], and MS2Query[3]. Furthermore, mass spectral similarity scores, like the cosine score, modified cosine score[4], Spec2Vec[5], MS2DeepScore[6], and others[7–9], play a crucial role in in silico annotation and organization approaches like library matching, analog searching, and organizing spectra by molecular networking.

The most widely used classical spectrum similarity measure, the cosine score, evaluates similarity based on visual equivalence in fragmentation patterns, making it effective for identifying (near-)identical molecules under strict conditions. The so-called modified cosine score considers both neutral losses and direct matching fragments during signal alignment. Thereby, it can account for a single structural modification and is used for searching structurally similar molecules[4,10,11]. Nevertheless, both scores fail to serve as general proxies for chemical similarity, as they struggle to account for more complex fragmentation relationships arising from multiple structural modifications[5]. Additionally, both metrics assume similar experimental conditions, making them sensitive to variations in ionization mode, instrument type, collision energy, and data processing pipelines. Consequently, they

[1]Bioinformatics Group, Wageningen University & Research, Wageningen, the Netherlands. [2]The National Phenome Centre, Section of Bioanalytical Chemistry, Division of Systems Medicine, Department of Metabolism, Digestion and Reproduction, Faculty of Medicine, Imperial College London, Hammersmith Hospital Campus, London, UK. [3]Institute of Organic Chemistry and Biochemistry of the Czech Academy of Sciences, Prague, Czechia. [4]mzio GmbH, Bremen, Germany. [5]Centre for Digitalisation and Digitality (ZDD), University of Applied Sciences Düsseldorf, Düsseldorf, Germany. [6]Department of Biochemistry, University of Johannesburg, Johannesburg, South Africa. [7]These authors jointly supervised this work: Justin J.J. van der Hooft, Florian Huber.
✉e-mail: niek.dejonge@wur.nl; justin.vanderhooft@wur.nl; florian.huber@hs-duesseldorf.de

identify only a tiny fraction of the dense chemical relationships found in complex samples[5,12].

Mass spectrometry can be performed in two ionization modes: positive and negative. How suitable a particular ionization mode is for detecting a metabolite largely depends on the metabolite's structure[13,14]. Consequently, mass spectrometry data are often acquired in both ionization modes to cover a larger fraction of the metabolome of the measured samples. While mass fragmentation spectra are highly similar for the same molecule when recorded in the same ionization mode with the same acquisition parameters, this is often not the case when comparing to a mass spectrum recorded in the other ionization mode[15]. By design, both cosine and modified cosine scores are, in these cases, not suitable to compare spectra across different ionization modes. As a result, positive and negative ionization mode mass spectra are mostly analyzed separately, for instance, by searching in separate reference libraries and creating two separate molecular networks[16–18]. Where approaches like MolNotator[19] and Ion Identity Molecular Networking[20] can merge positive and negative ionization mode spectra into one network, they require adduct identification based on well-aligned retention times and the recognition of specific mass differences between mass features. Achieving retention time alignment can be cumbersome and necessitates using the same chromatography column for both positive and negative ionization modes. A cross-ionization mode MS[2] similarity metric could alleviate these challenges by enabling streamlined computational workflows that align positive and negative ionization mode data.

To this end, we developed a mass spectral similarity metric that can predict chemical similarity between mass spectra of not only the same, but also different ionization modes. The approach for this similarity metric is based on the Siamese neural network architecture used in the previous version of MS2DeepScore[6]. The original MS2DeepScore model was able to predict chemical similarities with good overall accuracy, but it has several shortcomings. Firstly, separate models had to be trained for positive and negative ionization mode data, which meant less training data for each model and no cross-ionization mode applications. Secondly, the former MS2DeepScore models were trained on MS[2] fragments only; however, spectral metadata like precursor $m/z$, ionization mode, or other acquisition parameters could become valuable information for improving prediction quality. In the present work, we explore and evaluate the addition of metadata to the model input and show that using ionization mode and precursor $m/z$ as input improves model performance. Thirdly, we introduce a pair sampling algorithm that reduces biases introduced during model training. Finally, we introduce a method that can estimate the mass spectral embedding quality for each input spectrum. This allows users to filter out spectra for which the MS2DeepScore predictions are unreliable, e.g., due to low spectral quality or when spectra differ substantially from the training data. This further improves the reliability of MS2DeepScore results.

Our latest model is now also available in mzmine[21], seamlessly integrating MS2DeepScore molecular networking of thousands of spectra in seconds with mzmine's feature detection workflows and statistical analysis dashboards. The interactive network visualizer facilitates exploring the chemical space by combining multiple spectral similarity metrics. This local deployment offers scientists without programming expertise easy access to MS2DeepScore molecular networks. In addition to the above-mentioned key aspects, this work also contains hyperparameter optimization, which leads to better chemical similarity prediction. In addition, we added a training pipeline that makes training models easier, more streamlined and more robust.

In this work, we develop a model that predicts chemical similarity between mass spectra of not only the same, but also different ionization modes. We demonstrate the utility of this model through three case studies, showcasing its ability to integrate positive and negative ionization mode spectra into a unified analysis. In these case studies, the model correctly connects cross-ionization mode spectra with high chemical similarity, thereby finding more chemical relations between metabolite features. We also highlight the possibility of directly visualizing MS2DeepScore embeddings using UMAP[22] and provide an interactive plot for intuitive exploration of both positive and negative ionization mode spectra of the case study. Annotation by experts confirmed the validity of the discovered links between positive and negative ionization mode spectra. By enabling cross-ionization mode molecular networking and embedding visualizations, our model paves the way for extracting deeper insights from untargeted metabolomics data.

## Results

### Cross-ionization mode models

MS2DeepScore 2.0 models are trained on mass fragmentation spectra in both ionization modes (see Fig. 1). This resulted in a model that performs well for predicting chemical similarity between spectra acquired in the same ionization mode, but also for predicting chemical similarity between positive and negative ionization mode spectra. By using these predicted similarities by MS2DeepScore, both positive and negative ionization mode spectra can be visualized in one unified molecular network.

To test the performance of MS2DeepScore, a test set of annotated spectra is created, which is not used during the training of the model. The performance of the model is assessed by comparing the predicted chemical similarity by MS2DeepScore with the known true chemical similarity. Figure 2 shows the predicted MS2DeepScore scores between pairs of test spectra, showing a clear correlation between true chemical similarity and predicted chemical similarity. Figure 2b shows that even for predictions across ionization modes, there is a strong correlation between the predicted chemical similarity and the true chemical similarity.

### Case studies MS2DeepScore

The capabilities of cross-ionization mode MS2DeepScore models are illustrated with three case studies. One with human urine samples, one with human plasma samples, and a third using samples of *Rumex sanguineus*, a wild edible plant. For both the human urine sample and human plasma sample, molecular networks were formed by using the predicted similarity by MS2DeepScore to create edges.

In Fig. 3a, multiple cross-ionization mode networks were formed linking biochemically relevant metabolites together in the urine case study. The highlighted clusters in Fig. 3a have been manually curated and annotated by experts. In total, 37 spectra were manually annotated, resulting in the annotation of the structural identity of 13 different metabolites. The confidence level[23] of the annotations can be found in Supplementary Table 1. For example, the left cluster contains caffeine-related molecules, all part of known caffeine metabolism pathways[24]. This shows that MS2DeepScore was able to highlight cross-ionization mode connections in the molecular network that correspond to real metabolic pathways. The results of the molecular network formed for human blood plasma samples are shown in Supplementary Fig. 25.

MS2DeepScore also enables alternative visualization methods that overcome some limitations of molecular networking. One limitation of molecular networking is that it does not depict relationships between separate clusters. A second challenge with molecular networking is choosing a hard cut-off for when connecting an edge. If this threshold is set too high, relevant connections are not depicted, while if this threshold is set too low, a hairball of connections can form, which makes it challenging to interpret the data. An alternative visualization method is using UMAP to visualize MS2DeepScore embeddings, which overcomes both of these limitations of molecular networking. Each datapoint in the UMAP represents one spectrum, and spectra located closely together in the UMAP have high predicted chemical similarity. As an intermediate output, MS2DeepScore produces mass spectral

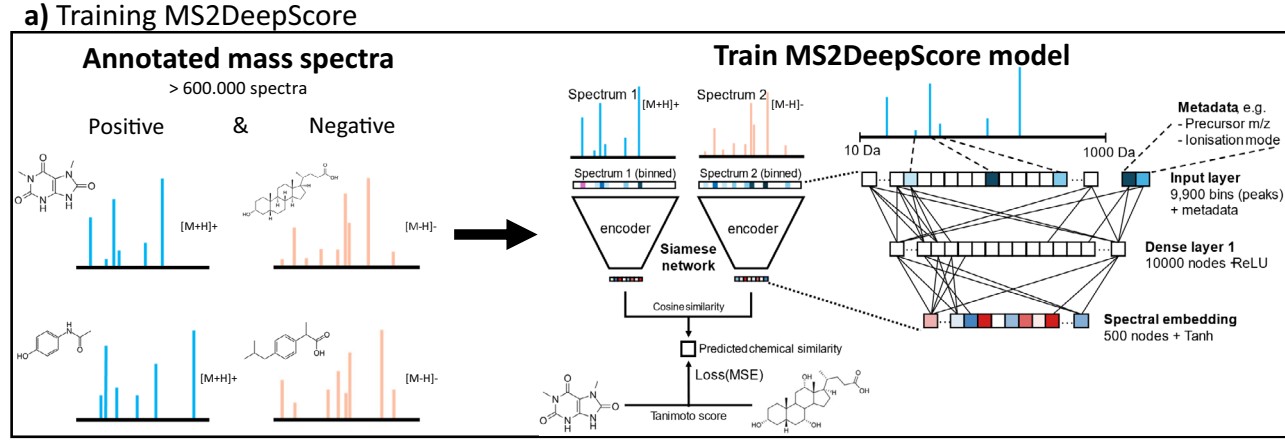

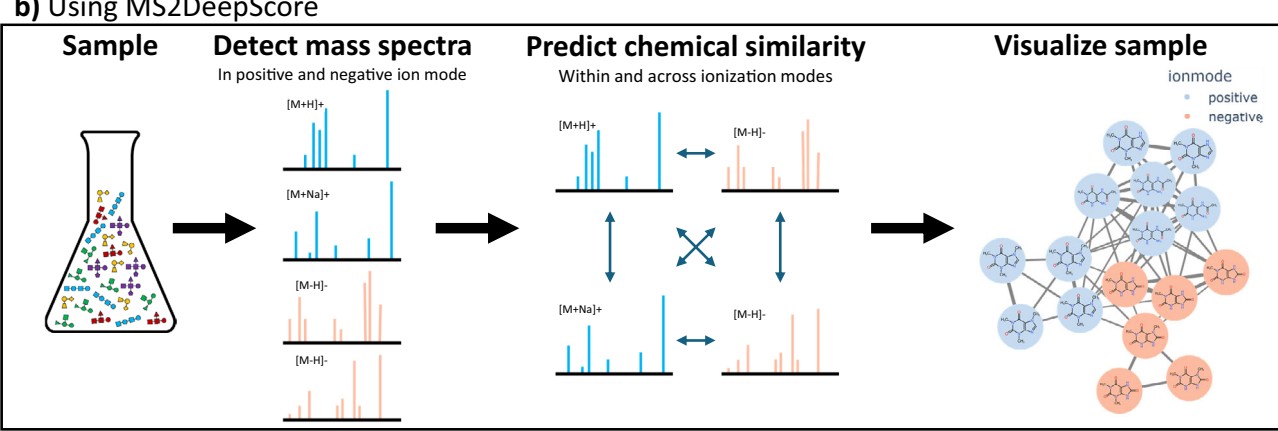

**Fig. 1 | Schematic overview of model training and intended use. a** Training MS2DeepScore. MS2DeepScore is trained on annotated mass spectra from public libraries. The model is trained to predict chemical similarity between pairs of mass spectra. By training the model on both ionization modes at the same time, the model is trained to predict chemical similarity within and across ionization modes. Models can be trained with additional in-house annotated mass spectra to further improve model accuracy. A Siamese neural network architecture is used. The input layer comprises scan metadata and fragment data after binning the *m/z* axis and applying square root transformation to the signal intensities. A single dense layer converts the input to a numerical vector (embedding) of length 500. The model is trained to create embeddings for which the cosine similarity between two embeddings correlates well with chemical similarity (Tanimoto score). **b** Using MS2DeepScore. MS2DeepScore predicts chemical similarity both within and across ionization modes. By using these predicted similarities to define edges, a unified molecular network can be constructed that connects spectra from both ionization modes. Visualizing the chemical space of a sample helps with data exploration, prioritization of spectra in clusters of interest and annotation propagation. Blue indicates positive ionization mode spectra, and orange indicates negative ionization mode spectra.

embeddings. UMAP can plot these 500-dimensional embeddings in a 2D representation of the chemical space. Figure 3b shows the UMAP[22] representation of the embeddings of the *Rumex sanguineus* case study. In this UMAP representation, we combined both positive and negative ionization mode spectra into a single representation. We observe mixing of the embeddings of both ionization modes in 2D space and highlight examples of annotated structures. To highlight the difference between positive and negative ionization mode spectra, we have highlighted an example of a positive and negative mode spectrum of Rutin. Both spectra have a similar embedding and are therefore visualized closely together in the UMAP representation, indicating high predicted chemical similarity. The spectra of this molecule in different ionization modes are visually different, but still MS2DeepScore 2.0 was able to predict chemical similarity of 0.7, while the modified cosine score results in a similarity of 0.3 and the cosine score returns a similarity of 0.0. The full UMAP representation of the embeddings is available as an interactive HTML file, allowing for exploring the case studies further, see "Data availability" section.

### Uncertainty evaluation
MS2DeepScore predictions can be unreliable for some mass spectra. This could, for instance, be due to bad or incomplete fragmentation,

fragments of multiple metabolites in one spectrum (i.e., "hybrid" spectra), or simply because there were no similar spectra in the training data. Here, we have developed and assessed our Embedding Evaluator model. This model predicts the mean squared error (MSE) from the embedding of a spectrum, see Fig. 4a. Figure 4b, shows that there is a strong correlation between predicted MSE and real MSE. Figure 4c shows the effect of removing the spectra with the highest predicted MSE. By filtering out spectra that have a high predicted MSE, the prediction accuracy between the remaining spectra increases. Additional analysis in Supplementary Figs. 21 and 22 tested the correlation between predicted MSE and features like the number of fragments, precursor *m/z*, ionization mode, and signal intensities. This showed some clear trends, like a higher predicted MSE for spectra with a low number of fragments and a higher predicted MSE for smaller metabolites.

### Additional metadata input
MS2DeepScore 2.0 allows for adding metadata as input to the neural net. Experiments with using additional metadata showed that adding precursor *m/z*, ionization mode, and adduct type as input for the model notably improved the performance, while one-hot encoding of the instrument type did not (see Supplementary Fig. 5). Considering

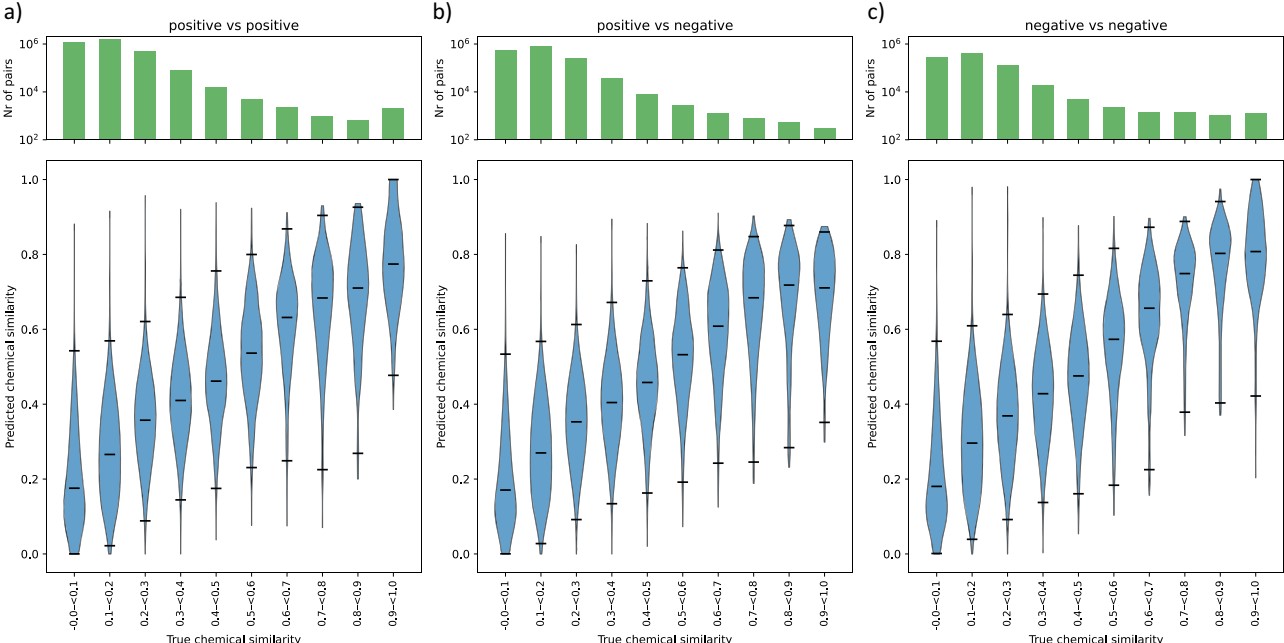

**Fig. 2 | Dual-ionization mode MS2DeepScore model predicts chemical similarity between and across ionization modes.** A test set of 32,052 spectra is used, which were not used to train the model. Predictions are made between all test spectra, followed by taking the average per unique molecule pair. The violin plots show the kernel density estimation (KDE) of the predicted values, the black lines represent the median and the 1st and 99th percentile for each bin. The dashed grey line indicates the y = x line. The bar plot on the top shows the log-scaled count of the number of unique molecule pairs in each bin with the corresponding chemical similarity. The metric used for chemical similarity prediction is the Tanimoto score between Daylight fingerprints. The raw data and notebook required to reproduce this figure is available in the Source Data file. **a** Predictions between pairs of positive ionization mode spectra. **b** Predictions between pairs of positive and negative ionization mode spectra. **c** Predictions between pairs of negative ionization mode spectra.

that the adduct type remains unknown or is annotated with lower accuracy in many common mass spectrometry workflows, the model used in the main text relies only on precursor $m/z$ and ionization mode as metadata.

### Sampling algorithm
One of the key challenges in training a model to predict Tanimoto scores is the highly non-uniform distribution of these scores across possible molecule pairs. Low Tanimoto scores are several orders of magnitude more frequent than high Tanimoto scores. The sampling algorithm is optimized to result in balanced sampling over Tanimoto scores and equal sampling of the different molecules. With this developed sampling algorithm, we achieve a good balance of molecule sampling frequencies with a maximum of <15% difference between molecules (see Supplementary Fig. 8d) and an exactly equal sampling frequency over the whole Tanimoto score range grouped in 10 bins. More details about the sampling algorithm optimization can be found in the Supplementary Note 2.

### Comparison to the original MS2DeepScore model
This paper introduces multiple advancements to the original MS2DeepScore paper, including adding metadata and the optimized sampling algorithm. To compare the performance, the original MS2DeepScore model (version 0.2.0) was retrained and tested on the same training and test set used here. Supplementary Fig. 14 shows side by side violin plot, directly comparing the performance of the original MS2DeepScore model to the developed MS2DeepScore model. Supplementary Fig. 15 shows the average MSE per Tanimoto bin, showing that the performance is improved by the training approach introduced here.

### Comparison with single ionization mode models
MS2DeepScore 2.0 has the capability to predict chemical similarity across ionization modes. However, it is important that the within-

ionization-mode prediction accuracy is not affected too much by training on both ionization modes at the same time. The comparison to the original MS2DeepScore model in Supplementary Figs. 14 and 15 already shows an improved performance for the dual ionization mode model. However, this comparison also includes the here introduced improvements to the model architecture and the inclusion of precursor $m/z$ as input to the model. To only test the effect of training a model on both ionization modes at the same time, two additional models were trained on a single ionization mode using the same hyperparameters as the dual ion mode model. The results are summarized in Supplementary Figs. 16, 17, and 18. Supplementary Fig. 18 compares the MSE for the within-ionization-mode predictions for a model trained on both ionization modes and a model trained on only a single ionization mode.

Models trained on only positive ionization mode spectra or on spectra of both ionization modes show comparable within-ionization-mode performance. The model trained on only negative ionization mode spectra, however, results in lower losses for lower Tanimoto bins, but a higher loss in the 0.9-1.0 Tanimoto bin when compared to the dual-ionization-mode model. Supplementary Fig. 17 shows a side by side violin plot, enabling direct comparisons between the distributions of predicted scores.

### Speed of training MS2DeepScore
To achieve fast training and prediction times, the entire model was implemented using Pytorch[25]. The MS2DeepScore 2.0 model was trained in 11.2 h on a server with Intel Xeon gold 6342 2.8 Ghz, Nvidia A40 GPU, and 512 GB Memory.

## Discussion
MS2DeepScore is able to make reliable predictions between mass spectra measured under different conditions, even if hardly any of the fragments overlap. We show that an MS2DeepScore model trained on

**a)** Molecular network urine case study

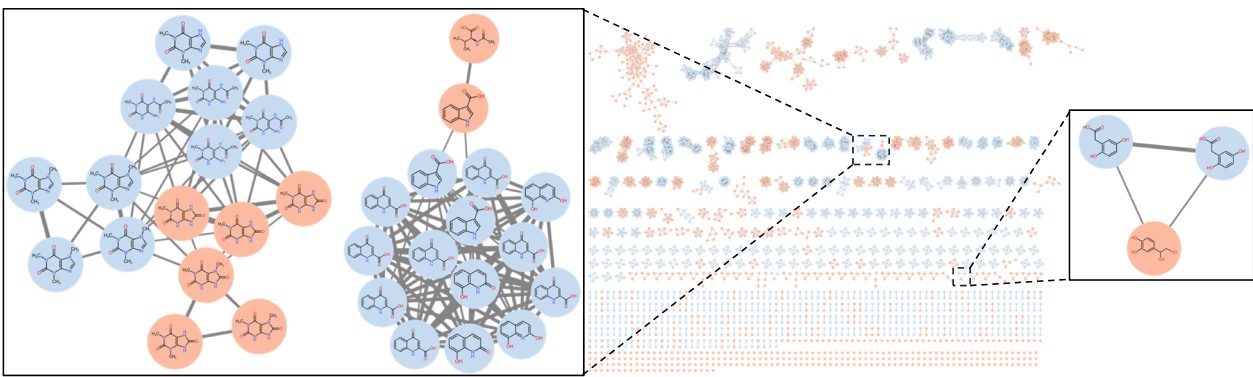

**b)** UMAP embedding representation *Rumex sanguineus* (plant) case study

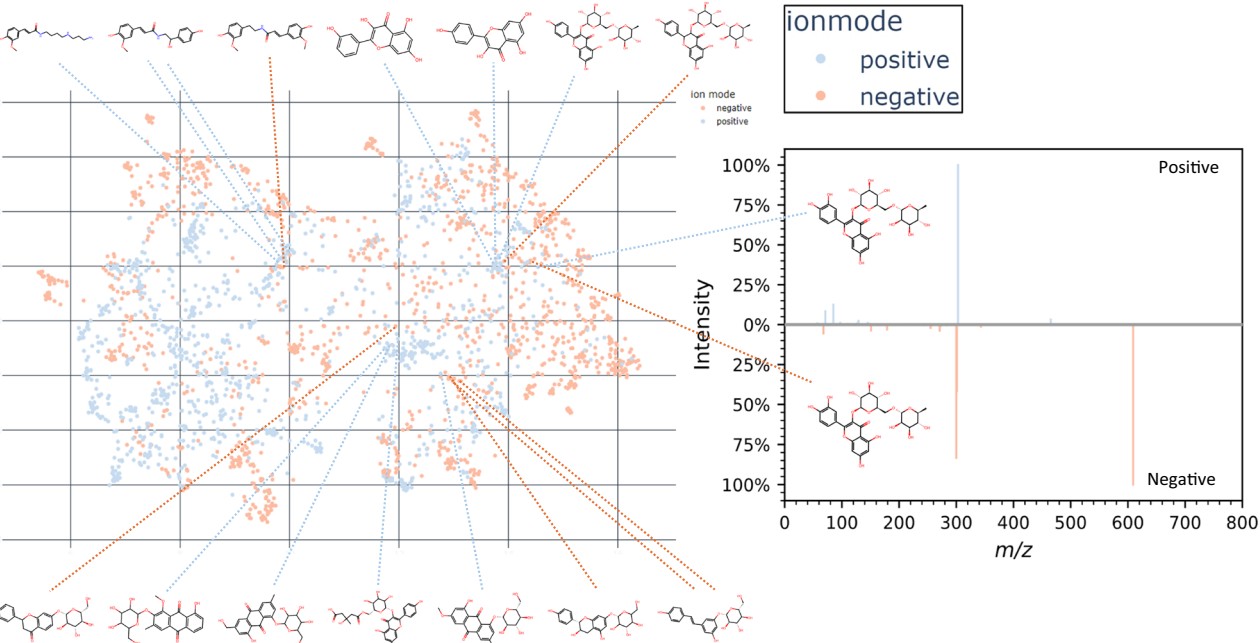

**Fig. 3 | Visual sample representations created with MS2DeepScore cross-ionization-mode model predictions.** By predicting chemical similarity between both the positive and negative ionization mode spectra, spectra of both ionization modes can be visualized together. The raw data and notebook required to reproduce this figure are available in the Source Data file. **a** A molecular network of the urine case study created by using MS2DeepScore similarity scores. An edge is created for an MS2DeepScore larger than 0.85. We highlight a few examples where MS2DeepScore was able to predict close chemical similarity between positive and negative ionization modes. The orange nodes correspond to negative ionization mode spectra and blue nodes correspond to positive ionization mode spectra. Spectrum mirror plots of all cross-ionization mode pairs visualized in the molecular network can be found in Supplementary Fig. 26. Spectrum mirror plots for exact matches across ionization modes are available in Supplementary Fig. 27. An interactive version of the molecular network can be loaded in Cytoscape, the data is available via the "Data availability" section. **b** UMAP representation of the MS2DeepScore 2.0 embeddings of the *Rumex sanguineus* case study. Each dot represents a spectrum and closely positioned spectra have high predicted chemical similarity. The molecular structure for multiple annotated mass spectra is visualized as examples. Two spectra are highlighted, which both correspond to the same molecule, but were recorded in positive and negative ionization modes. MS2DeepScore 2.0 correctly predicted very similar embeddings, while the fragments do not overlap. An interactive version of the full UMAP plot is available as an HTML file, see "Data availability" section.

both ionization modes can predict good estimates of the chemical similarity between spectra measured in different ionization modes (Fig. 2b). In addition, the accuracy for within-ionization-mode predictions has also improved compared to the previously published MS2DeepScore model[6].

Spectral similarity scores like MS2DeepScore are used for various downstream applications; however, we recognize two main ones: (1) organizing and visualizing samples by identifying high chemical similarity within a dataset, and (2) library searching for exact matches or structurally related analogs. The here introduced cross-ionization-mode prediction functionality primarily benefits the first task. Our case studies show that MS2DeepScore correctly predicts multiple high similarities between positive and negative ionization mode spectra. These edges correctly bridge clusters that were previously disjoint when using only within-mode predictions.

However, an important limitation remains: the number of cross-ionization-mode edges that exceed a similarity score of 0.85 and fall within the top-10 highest scores is low. Lowering the threshold for the minimum predicted similarity score increases the number of cross-ionization-mode connections, but this is not generally recommended, since this also substantially increases the number of false positives. This then leads to the molecular network becoming a hairball, hindering effective data analysis. These limitations are less consequential in the UMAP-based visualization, where no strict threshold is required.

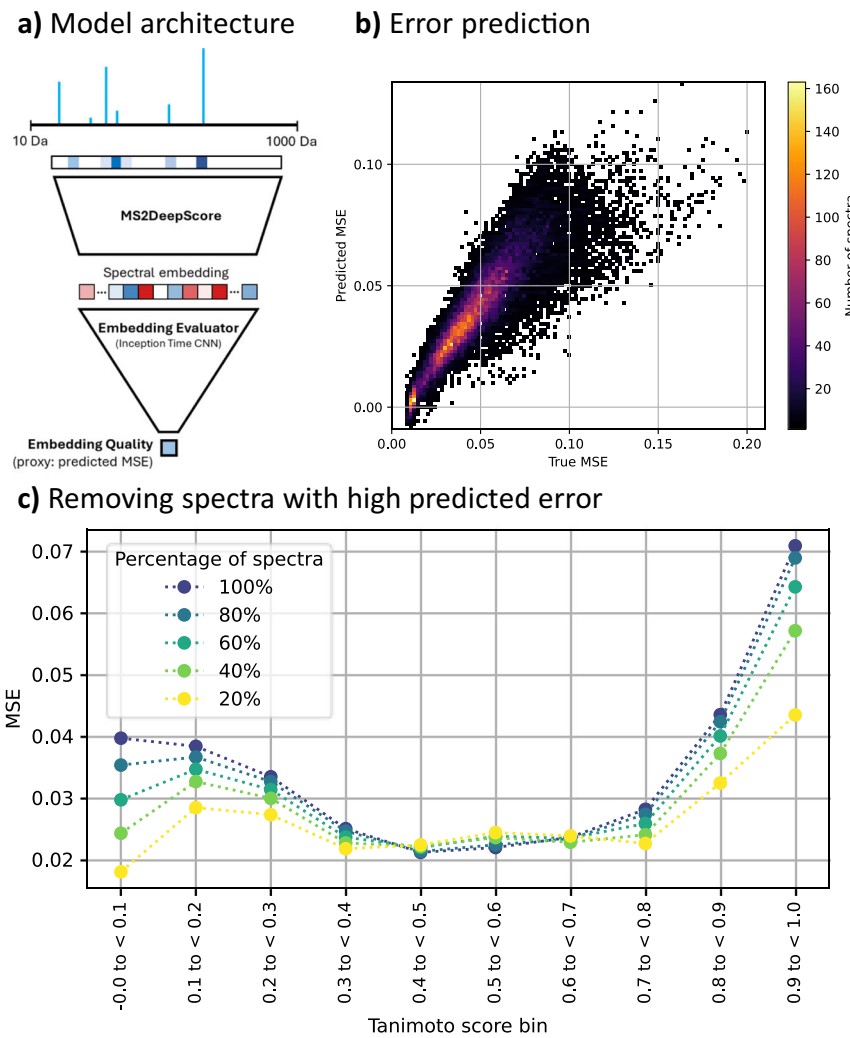

**Fig. 4 | Results embedding evaluator.** The embedding evaluator predicts if MS2Deepscore can make reliable chemical similarity predictions for an input spectrum. This embedding evaluator is trained to predict the mean squared error (MSE) for a spectrum. The raw data and notebook required to reproduce this figure are available in the Source Data file. **a** Architecture for the embedding evaluator. **b** The predicted MSE is plotted against the true MSE for each spectrum in the test set. The test set contains 32,052 spectra. **c** Different thresholds are used to remove spectra from the test set with high predicted MSE. This results in 5 subsets of spectra with respectively 20, 40, 60, 80, and 100 % of the test spectra retained. The plot shows the real MSE per bin for predicting chemical similarity for the selected subsets of the test set. This shows that the accuracy of MS2DeepScore improves if the Embedding Evaluator is used to remove spectra with high predicted MSE. The MSE is calculated by taking the average MSE between all spectra of two molecules, followed by taking the average per Tanimoto bin.

---

Here, spectra from positive and negative ionization mode spectra are integrated smoothly into a unified chemical space, without the need for specific thresholds.

The cross-ionization mode capability is the first step to enable cross-ionization mode library searching for analogs in the future. However, when implemented in a naïve approach of selecting the top 1 highest candidate, there is no added benefit from using a cross-ionization mode model (See Supplementary Note 10). There is only a small number of cases where the predicted score is both above the threshold of 0.85 and higher than all the within-ion-mode connections. Our previous work for MS2Query showed that the accuracy of analog searching using chemical similarity prediction tools like MS2Deep-Score can be improved substantially by reranking the best candidates based on the similarity predictions for multiple closely related library molecules[3]. We therefore recommend an approach like that introduced in MS2Query as a starting point to enable reliable cross-ionization mode analog searching. At present, the principal value of cross-ion-mode MS2DeepScore predictions lies in within-sample organization and visualization across ionization modes, where the method already provides clear and actionable benefits.

Other methods like Ion Identity Molecular Networking[20] and MolNotator[19] allow users to combine mass spectra of different adducts that belong to the same metabolite by using mass difference and retention time alignment. MolNotator applied this principle to merge molecules across ionization modes. A limitation of these methods is that only metabolites can be linked across ionization modes if two mass spectra were recorded for the same metabolite in both the positive ionization mode and the negative ionization mode. Since MS2DeepScore can also predict chemical similarity across ionization modes if two metabolites are not identical, additional connections can be made between highly similar molecules across ionization modes. However, a limitation of MS2DeepScore is that no clear distinction is made between identical metabolites and closely related metabolites across ionization modes. Future work could combine approaches like Ion Identity Molecular Networking with MS2DeepScore to connect and confirm identical matches across ionization modes and to clearly distinguish closely related molecules from identical molecules.

In this work, we made substantial updates to the MS2DeepScore pipeline. For example, the pair sampling algorithm to train MS2DeepScore was optimized. Sampling pairs during training is a

crucial step in training MS2DeepScore, since low Tanimoto scores are orders of magnitude more frequent than high Tanimoto scores. Here, we introduce a sampling algorithm that not only balances the sampled pairs over equally spaced Tanimoto bins but also balances the sampling frequency of each molecule and even the distribution of the Tanimoto scores per unique molecule. This pair sampling algorithm reduces potential biases in the training data and ensures that the diversity in the training data is used well. Whilst these changes were a substantial improvement in making good use of the chemical diversity in the training set, we do note that our sampling algorithm did not enforce balanced sampling of the different ionization mode pairs. Because more spectra were available in positive ionization mode (519,580) than in negative ionization mode (145,594), this resulted in sampling more positive ionization mode pairs compared to negative ionization mode pairs. Having more balanced sampling over the ionization modes might improve model performance for negative vs. negative ionization mode predictions. In addition, since there are differences in the Tanimoto score distributions for positive and negative ionization mode spectra, this might result in not having an equal number of pairs per Tanimoto bin for each ionization mode. For example, since there were not many high Tanimoto score examples between positive and negative ionization mode spectra during training, it is likely that this explains the observation that almost no predictions above 0.9 were made for cross-ionization mode pairs. In future work, the sampling algorithm could be further optimized to also enforce balanced pair sampling for the different ionization mode pairs.

In addition, allowing metadata as input to the model improves performance, see Supplementary Fig. 5. The model used in the main text uses precursor *m/z* and ionization mode as metadata input. Using the adduct as input was also beneficial for model performance. There are methods to predict adduct information from MS[1] (full) scans[26]; however, not all preprocessing tools generate reliable adduct information. Therefore, we decided not to include this in the default model, since using incorrect adduct information could result in reduced performance. Currently, the model does not use MS[1] data directly. In future work, it would be interesting to include MS[1] data in the training of MS2DeepScore models. However, a challenge is that public mass spectral libraries generally do not have annotated raw MS[1] spectra available. Directly including MS[1] data, or alternatively predicted adducts or molecular formulas as features into the model, could potentially further improve MS2DeepScore performance.

MS2DeepScore models are trained to predict chemical similarity scores, using the Tanimoto coefficient between Daylight fingerprints as the primary metric. Widely regarded as effective for fingerprint-based comparisons, the Tanimoto score has become a standard in cheminformatics applications[27–29]. However, molecular similarity is inherently subjective, varying by context and application[30,31]. Even when we restrict ourselves to fingerprint-based metrics, many possible variants with different strengths and weaknesses exist[32,33]. Future work could explore alternative similarity metrics and fingerprints, leveraging the flexible architecture of MS2Deepscore to expand its applicability across diverse tasks.

The cross-ionization mode MS2DeepScore model uses a Siamese neural network architecture similar to the original MS2DeepScore paper[6]. Hyperparameter optimization resulted in using a dense network of a different size. For benchmarking, we picked the best-performing model. Smaller models are possible with only slight performance reductions. For details, see Supplementary Figs. 2, 3, and 4. If computing time or embedding size is crucial for an application, an MS2DeepScore model with a smaller model architecture or embedding size can easily be trained.

Fragment signals are initially binned, which might result in the loss of useful information. In Supplementary Fig. 6 we show that using bins smaller than 0.1 Da did not result in improved performance. However, exploring alternative methods that do not require binning could potentially further improve model performance. For instance, the recent DreaMS model uses a transformer architecture, which does not require binning of spectral data[34]. By pretraining a DreaMS model on unsupervised data, followed by transfer learning to create a spectral similarity prediction model, it is possible to train a model that works without binning. So far, the DreaMS model has only been trained on positive ionization spectra, making it unsuitable for training a cross-ionization mode model. In future work, it would be valuable to attempt pretraining models on both ionization modes, followed by transfer learning to create a chemical similarity predictor. For future work exploring deep learning algorithms, we recommend building on the groundwork done in MS2DeepScore 2.0 by reusing the implemented sampling algorithm, automated training pipeline, and benchmarking methods to make the results reproducible and comparable.

The prediction quality of the MS2DeepScore model is sensitive to the quality and type of the input spectra. Poor predictions are expected for low-quality spectra with limited fragmentation, chimeric mass spectra from multiple precursor ions, or spectra with little similarity to our training data. In the original MS2DeepScore paper[6], the uncertainty was estimated using a Monte-Carlo dropout regularization[35]. In later real-world applications, however, we noted that this was a subpar solution. For example, we noticed that spectra with little similarity to the training data, as well as low-quality spectra, often received very similar embeddings. This is very detrimental because similar embeddings will lead to mostly false predictions of high chemical similarities. Hence, MS2DeepScore 2.0 is now complemented by an uncertainty estimation for individual input spectra. This model predicts the MSE from a spectral embedding created by MS2DeepScore. We demonstrated that the overall accuracy can be raised by removing the test spectra with a high predicted uncertainty, see Fig. 4. We anticipate that other tools can use this method for uncertainty and spectral quality estimation.

Given the enormous range of possible mass spectral datasets and applications, there remains a risk of our model not being well-suited for very specific tasks or chemical classes. In such cases, we recommend training a custom MS2DeepScore model. Training a MS2DeepScore model is now relatively easy since an automatic training pipeline is available. For smaller custom datasets, we recommend merging them with larger available datasets, such as the here-used public libraries, before training a model from scratch. We speculate that a promising alternative route could be to start with our pre-trained model and run additional training on the custom reference data, a commonly used "fine-tuning" strategy in deep learning.

MS2DeepScore is available through PyPI via pip, is actively maintained and adheres to best practices in software development. Most code is covered by unit tests and supported by a continuous integration (CI) pipeline to ensure reliability and robustness. To make MS2DeepScore accessible to a wider audience, which is unfamiliar with using basic Python, MS2DeepScore is now also available in mzmine[21]. Within mzmine, MS2DeepScore-based molecular networking can be combined with feature detection, metabolite annotation, and interactively linked to statistical analysis. We anticipate this will make MS2DeepScore and cross-ionization mode predictions available to a wide audience of chemists without programming experience.

The ability to reliably predict chemical similarities across ionization modes creates entirely new options for mass spectral data exploration by combining positive and negative ionization mode data. Similarity-based graphs can now be generated independent of the ionization mode, rendering cross-ionization mode molecular networking feasible. Furthermore, our model can be used as a basis to use the larger positive ionization mode reference spectral library as a source for annotation of the negative ionization mode data, and vice versa. We expect that this will help researchers to more quickly and comprehensively identify molecules and to make chemical and biological discoveries.

## Methods

### Metadata as input

MS2DeepScore 1.0 uses mass fragments as an input to predict chemical similarity between mass spectra[6]. In the current work, MS2DeepScore 2.0 allows for the use of additional metadata of the fragmentation spectra. This is implemented in a flexible way, which allows adding any type of metadata as an input into the model. Numerical data, e.g., precursor *m/z* or collision energy, is transformed to values closer to 1, to have input in a similar order of magnitudes, to optimize training. Textual inputs, like ionization mode or instrument type, are one-hot encoded. For the dual-ionization mode model used in the main text, precursor *m/z* and ionization mode were used as additional metadata input. As a part of the current study, other experiments were also run with instrument type and/or adduct type as additional input(s). An overview of the selected model architecture can be found in Fig. 1.

### Tanimoto score

As a metric for chemical similarity between two molecules, the Tanimoto score between molecular fingerprints was used[36]. An RDKit[37] daylight-like binary fingerprint (4096 bits) was generated for each unique 2D structure. This Tanimoto score was used for training and benchmarking and will be referred to as the Tanimoto score. The original MS2DeepScore model used 2048 bits. Supplementary Fig. 7 shows the difference in MSE when training and benchmarking a model on 2048 or 4096 fingerprint bits. Since the resulting accuracy is different for different fingerprint lengths, this makes direct comparisons of benchmarking results in the figures of the original MS2DeepScore paper impossible.

### Spectrum pair selection for training

One of the key challenges in training a model to predict Tanimoto scores is the highly non-uniform distribution of these scores across possible molecule pairs. Low Tanimoto scores are several orders of magnitude more frequent than high Tanimoto scores. In our previous work[6], this imbalance was partly mitigated by a data generator that selected a molecule pair belonging to a random Tanimoto score bin for each pair selection step. However, molecules in the used dataset often lacked partners in the high Tanimoto score ranges. As a result, even though the former data generator substantially reduced the bias, there was still a considerable shift towards lower Tanimoto scores. In addition, the selection of the second molecule in a pair was not equally distributed, leading to high variability in sampling frequency per unique molecule.

To address these issues, we developed a pair sampling algorithm optimized for balanced sampling across Tanimoto score bins and near uniform sampling frequencies for each unique molecule. During training, the pair sampling algorithm loops over selected molecule pairs and randomly selects two corresponding spectra per molecule pair, because often multiple spectra are available for a single molecule. In this work, two molecules were considered the same if the first 14 characters of their InChIKeys were equal, thereby ignoring stereochemistry. The pair sampling algorithm loops multiple times over the selected set of molecule pairs, but the corresponding spectra are randomly resampled every loop.

Before training the model, a balanced set of molecule pairs was selected. The molecule pair sampling happened per Tanimoto bin, but during the sampling, the molecule sampling count was tracked. The sampling algorithm started by selecting the least frequently sampled molecule with pairs available in the Tanimoto bin. From the candidate pairs for this molecule, the second molecule with the lowest sampling count was chosen. Resampling of molecule pairs was allowed, enabling both a close-to-equal sampling frequency across unique molecules and a balanced distribution over Tanimoto bins. To minimize resampling, the algorithm prioritized the least sampled available pairs before

selecting the least sampled second molecule. This sampling algorithm significantly improved sampling balance. However, some molecules were still sampled up to six times more than others. To further reduce this imbalance, a maximum sampling count per molecule was introduced, limiting sampling frequency disparities to less than 15%. Details of the experiments conducted to optimize the sampling algorithm are provided in Supplementary Note 2.

### Binning spectra

Before training, the fragments were binned to make them suitable as input for the neural network. Binning happened by making bins of 0.1 Da between $10 \leq m/z < 1000$ Da, resulting in 9900 bins. In the former MS2DeepScore work, bins were only included if they had at least one fragment in the training data. Instead, MS2DeepScore 2.0 uses all bins, even if none of the training spectra have a fragment in this bin. This reduces code complexity and reduces the risk of accidental mismatch between the binning method and model versions. Intensity values were transformed by square-root to reduce the impact of high intensity signals.

### Architecture improvements

MS2DeepScore 1.0 is implemented in Tensorflow[6,38]. Here, the entire MS2DeepScore 2 model was reimplemented using Pytorch[25]. This improved compatibility with GPUs and Apple M1 chips, but also overall code readability. A pipeline is now available that performs all steps necessary for training MS2DeepScore models. The wrapper function only requires a file with annotated mass spectra and the settings for model training. First, mass spectra are separated on ionization modes and split in test, train, and validation sets. After that, the model is trained, and benchmarking figures are created.

### Model settings

The original MS2DeepScore paper used two layers of 500 nodes with an embedding size of 200. Given the expanded training library and the dual-ionization mode training, it was expected that a different model architecture could result in better performance. Hyperparameter optimization was performed to determine an optimal configuration, as detailed in Supplementary Note 1. The final architecture consisted of a single layer with 10,000 nodes and an embedding size of 500, which was used for all models presented in the main text.

Compared to the former MS2DeepScore models, several other adjustments were made. The final layer activation function was changed from ReLU to Tanh[39], dropout and batch normalization were removed, and the settings for data augmentation were changed: augment removal max was changed from 0.3 to 0.2, augment intensity was changed from 0.4 to 0.2, and augment noise intensity was changed from 0.01 to 0.02. The exact settings were added as a JSON file to the Zenodo entry, see "Data availability" section.

### Input data filtering and splitting

For training the models, we combined multiple public libraries: the GNPS library[4], the MassBank EU library, the MassBank of North America (MoNA) library[40], and $MS^2$ spectra of MS$^n$Lib created by Brungs et al.[41]. After combining these libraries, they were first cleaned using the matchms library cleaning pipeline[42,43]. The settings for cleaning can be found in Supplementary Settings 1. Experiments that assessed the model performance for different minimum signal numbers and intensity thresholds can be found in Supplementary Fig. 1. After cleaning, the library consisted of 36,638 unique molecules and 519,580 spectra in positive ionization mode and 18,480 unique molecules and 145,594 spectra in negative ionization mode. A molecule was considered unique if the first block of its InChIKey identifier (14 letters) was equal, thereby ignoring stereochemistry.

The cleaned spectrum library was split by ionization mode and divided into training, validation, and test sets. We selected 1/20th of

unique molecules for both the validation and test sets, all corresponding spectra to these molecules were removed from the training set. For the positive and negative mode set, the selection of InChIKeys to use for the validation and test sets was different, since we might otherwise have introduced a bias in the validation set for spectra that were available in both ionization modes. The dual-ionization mode library was trained by combining the positive ionization mode training spectra and the negative ionization mode training spectra. The validation spectra were used for all experiments for the optimization of our model, like changing the filtering of input spectra, or adjustments to the model size. The test set was not used during any experimentation or hyperparameter optimization and was only used for benchmarking of the final model.

### Embedding and score uncertainty estimation

The prediction quality of the MS2DeepScore model is sensitive to the quality of input spectra and the similarity to the training data. To detect spectra that are hard to predict for MS2DeepScore, we designed a pipeline using a convolutional neural network that predicts the quality of a spectrum embedding. The "Embedding Evaluator" model is implemented using an Inception Time architecture[44] using Pytorch[25]. The embedding evaluator is trained to predict the mean squared error (MSE). For each training spectrum, the MSE is calculated by randomly sampling 999 other training spectra and calculating the MSE over these 999 pair predictions. The model is trained to predict the MSE from the embedding of the spectrum. The conceptual idea here is that the Embedding Evaluator will learn to identify embeddings for low-quality or out-of-distribution input data. In later applications, the predicted embedding qualities can be used for uncertainty estimation.

### Benchmarking

The mean squared error (MSE) was used as a loss function, measuring the difference between the predicted and actual Tanimoto score. During training, a sampling algorithm ensured equal numbers of spectrum pairs in each Tanimoto bin and a balanced representation of molecule pairs. However, due to the lower number of available validation spectra, it is not suitable to use the same sampling algorithm for the validation spectra, since this would significantly reduce the number of pairs available for benchmarking. To obtain a representative MSE for the validation and test set, the average loss per molecule pair was calculated by averaging the losses of all available spectrum pairs for each molecule pair. The used mass spectral library often contains multiple mass spectra for one molecule, in some cases up to several hundred spectra for the same molecule. By taking the average loss per molecule pair, we ensured that the model's performance is not judged mostly on the performance of a few molecules with a high number of mass spectra. The average MSE per molecule pair was then used to calculate the average MSE per Tanimoto bin. Ten equally spaced Tanimoto bins between 0 and 1 were used. The final loss used was the average MSE over these 10 bins. In addition to this benchmarking we analysed the performance of MS2Deepscore for different adducts and different compound classes, these results can be found in Supplementary Figs. 23 and 24. These analyses give insights into the general patterns of the strengths and weaknesses of the MS2DeepScore model. "Lipids and lipid-like molecules" and "nucleosides, nucleotides and analogs" had slightly better performance, while organic oxygen compounds had a higher error. Supplementary Fig. 24 shows that for the adducts [M+Na]+ and [M + K]+, the accuracy was lower than average. A comparison of MS2Deepscore to the modified cosine score is available in Supplementary Note 5.

### Case studies

To illustrate the possibilities MS2DeepScore 2.0 introduces, we show the capabilities of the model to create dual-ion-mode molecular networks and UMAP[22] embedding representations using three case studies with experimental datasets. The case studies are performed on human urine samples, human blood plasma samples and a *Rumex sanguineus* (plant) sample.

**Urine case study.** The MS[2] spectra used for this case study were acquired for urine fractions generated at the National Phenome Centre by reversed-phase liquid chromatography (RP-LC) as detailed in the supplementary methods and described by Albreht et al.[45]. Urine fractions were treated like intact urine samples and profiled using the previously reported UHPLC-RP-LC assay designed for small molecule separation on a Waters $2.1 \times 150$ mm (1.8 µm) HSS T3 column maintained at 45 °C[46,47]. LC-MS system was a Waters Acquity UPLC instrument coupled to Xevo G2-S TOF mass spectrometer (Waters Corp., Manchester, UK) via a Z-spray electrospray ionization (ESI) source. All LC conditions, the gradient elution program and the mass spectrometry parameters are detailed in the Supplementary Methods.

**Human blood plasma case study.** Lipid species present in the NIST Frozen Human Plasma standard reference material−SRM1950, were chromatographically separated using a UHPLC-RP-LC assay tailored for complex lipid separation on a $2.1 \times 100$ mm BEH C8 column (maintained at 55 °C) developed by Lewis, et al.[47]. For this analysis, LC-MS system was an ACQUITY Premier UPLC (Waters Corp., Milford, MA, USA) with a Premier solvent manager and column/heater modules and a H-Class sample manager and organizer coupled to a Xevo G3 QTof mass spectrometer (Waters Corp., Manchester, UK). All details of sample preparation, LC conditions, gradient elution program and mass spectrometry parameters are reported in the supplementary methods.

**Rumex sanguineus case study.** Metabolomics data from *Rumex sanguineus* were obtained from the study performed by Ramundi et al.[48]. Briefly, 25 mg of tissue was homogenized and extracted with a water/methanol/methyl tert-butyl ether mixture. Mass spectrometry data were acquired using an Agilent 1290 UHPLC coupled to a Thermo Q-Exactive Focus Orbitrap equipped with a heated electrospray ionization interface. Analysis was performed in both positive and negative ionization modes, with data-dependent acquisition (DDA). 24 samples were analysed, comprising 4 biological replicates extracted in duplicates (8 roots, 8 stems, and 8 leaves). The mass spectra from all samples were combined into a single file and processed with MS2DeepScore to create a cross-ionization mode UMAP visualization. Ramundi et al. annotated 347 metabolites, including both level 1 and level 2 annotations. These annotations are used to validate cross-ionization mode capabilities of MS2DeepScore, the molecular structures for some of the annotated spectra are visualized in Fig. 3b.

**Data preprocessing urine and blood plasma case studies.** MassLynx software (Waters, Manchester, U.K.) was used for data acquisition and visual inspection. The raw data files were converted from the Waters.RAW format to.mzML format using the msconvert tool from the ProteoWizard toolkit[49]. DDA files converted to.mzML format were peak picked and converted to.mgf format using MSDIAL ver.4.9.221218 Windowsx6434 using the following parameters for peak detection: min peak height = 1000 amplitude, mass slice width = 0.05 Da; MS2Dec: sigma window value = 0.5, MS[2] abundance cutoff = 200 amplitude; Alignment: RT tolerance = 0.05 min, MS[1] tolerance = 0.01 Da.

The peak-picked spectra were further processed by matchms[42,43]. For the urine case study, only MS[2] spectra were kept with more than four fragments, for the *Rumex sanguineus* and blood plasma case studies, no minimum number of peaks was set. The exact processing settings and logging can be found in the Jupyter notebooks on GitHub. The visualized structures in Fig. 2 and the annotated nodes in Supplementary Fig. 25 have all been manually annotated. In addition, putative annotations and analog predictions were done using MS2Query[3]. Annotations with a prediction higher than 0.7 are included in the interactive UMAP embedding visualization.

**Molecular networking**. In Fig. 3a and Supplementary Fig. 25, dual-ionization mode molecular networks are visualized with MS2DeepScore similarity edges and MS[2] as nodes. GraphML files were created using matchms. The graphml files were used for visualizing the molecular networks in Cytoscape[50]. The minimum MS2DeepScore cut-off used is 0.85, "top-n" is set to 20, meaning that only the top 20 highest-scoring similarity scores per spectrum were considered for creating edges. The link method used was mutual, which means only edges were added if the edge is in the top list of both nodes. For each node, the highest 10 scores that have a mutual link in the top 20 of bode nodes were used for creating an edge. These settings could still result in more than 10 edges connecting to a single node if an edge from another node was in the top 10 highest similarity scores with a mutual connection.

To highlight the capabilities of MS2DeepScore across ionization modes, we selected a few clusters that included both positive and negative ionization mode spectra. The clusters were manually annotated by experts. Supplementary Table 1 provides the method of annotation and confidence levels for all structures visualized in Fig. 3a.

**Embedding UMAP visualization**. MS2DeepScore generates spectral embeddings as intermediate output. These embeddings can be used directly to visualize spectra in 2D space by using dimensionality reduction methods. Here, we used UMAP to reduce the 500 embedding dimensions to two dimensions. The number of neighbors was set to 50; this setting influences how local or global the 2D representation is. The resulting UMAP representation can be found in Fig. 3b and is also available as an interactive plot, see "Data availability" section. The interactive plot can be colored based on ionization mode or ClassyFire[51] compound class annotation of MS2Query[3] analog predictions.

**Integration into mzmine**. MS2DeepScore is available through PyPI as a pip installable Python package. Even though little programming knowledge is required to apply MS2DeepScore models and clear tutorials are available, this was still a significant hurdle for scientists without programming experience. To offer an easy local deployment, MS2DeepScore has been integrated into mzmine[21], a modular MS data processing software. Now, mzmine enables Feature-based Molecular Networking (FBMN[52]) and Ion identity Molecular Networking (IIMN[20]) using MS2DeepScore in an interactive network visualizer coupled with compound annotation and statistics dashboards. This allows users to create molecular networks and explore the chemical space within the mzmine graphical user interface, without requiring command line or scripting. A tutorial for using MS2DeepScore within mzmine can be found here: https://mzmine.github.io/mzmine_documentation/module_docs/group_spectral_net/molecular_networking.html#algorithm-MS2DeepScore.

Integration of MS2DeepScore in mzmine required converting the MS2DeepScore model to the Torch script format, which is supported by the Deep Java Library (DJL). The package https://github.com/niekdejonge/MS2DeepScore_java_conversion contains scripts for converting existing MS2DeepScore models. The latest Torch script version of the MS2DeepScore model is available at https://doi.org/10.5281/zenodo.12628368 and can be automatically downloaded from within mzmine's molecular networking module. MS2DeepScore is available in mzmine starting from mzmine version 4.3.0.

### Reporting summary
Further information on research design is available in the Nature Portfolio Reporting Summary linked to this article.

## Data availability
The dual-ionization mode MS2DeepScore model and embedding evaluator model used in this study can be downloaded from Zenodo, https://doi.org/10.5281/zenodo.14290920. The training, validation, and test spectra can be downloaded from https://doi.org/10.5281/zenodo.13934470. All case study data can be found on https://doi.org/10.5281/zenodo.16311735. This includes an interactive version of the full UMAP plot as an HTML file and the required files to create the molecular networks in Cytoscape. Source data are provided with this paper. The raw data and notebooks required to reproduce each figure are available in the Source Data file. Source data are provided with this paper.

## Code availability
MS2DeepScore is available as PyPI package and therefore pip installable. The version used for this manuscript is version 2.5.3. All code is available on https://github.com/matchms/MS2DeepScore and version 2.5.3 can be downloaded from https://doi.org/10.5281/zenodo.18255344. The readme contains explanations on how to install and use MS2DeepScore and includes instructions for training an MS2DeepScore model on in-house data. The notebooks used for creating the benchmarking figures can be found in the folder https://github.com/matchms/MS2DeepScore/tree/main/notebooks/MS2DeepScore_2. The links to the raw data and notebooks required to reproduce each figure are available in the Source Data file. The mzmine source code for the PyTorch model integration via TorchScript format is available on the mzmine GitHub https://github.com/mzmine/mzmine/tree/master/mzmine-community/src/main/java/io/github/mzmine/modules/dataprocessing/group_spectral_networking/MS2DeepScore. The scripts for converting existing MS2Deepscore models to torchscript can be found on GitHub: https://github.com/niekdejonge/MS2DeepScore_java_conversion.

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

## Acknowledgements

The authors thank Corinna Brungs for sharing prereleases of the latest MS$^n$Lib library[41] and for her assistance with filtering out MS$^n$ and merged MS$^2$ spectra. The authors thank Tomáš Pluskal for hosting N.d.J. during a lab visit, which enabled the collaboration that led to the integration of MS2DeepScore in mzmine. N.d.J. thanks Dick de Ridder for helpful discussions and feedback on the results of MS2DeepScore 2.0. N.d.J. thanks the research lab of Soha Hassoun for valuable feedback on the first preprint. E.C. was supported by the Medical Research Council and National Institute for Health Research [grant number MC_PC_12025] and the Medical Research Council UK Consortium for MetAbolic

Phenotyping (MAP UK) [grant number MR/S010483/1]. Infrastructure support for E.C. was provided by the National Institute for Health Research (NIHR) Imperial Biomedical Research Centre (BRC). F.H. was supported by the Deutsche Forschungsgemeinschaft (DFG, German Research Foundation) [project 528775510].

## Author contributions

N.d.J. came up with the concept for cross-ionization mode similarity scores and wrote the first version of the manuscript. F.H. came up with the concept for the EmbeddingEvaluator. N.d.J., J.J.J.v.d.H., and F.H. designed the research and revised the manuscript. N.d.J., D.J., L.J.T., and F.H. contributed to the code. N.d.J. and F.H. designed and evaluated the code for the current version. N.d.J. and R.S. implemented MS2DeepScore in mzmine. E.C. annotated the case study data. Data from the urine samples used in this study were originally generated by the National Phenome Centre, Imperial College London. All authors contributed to the data analysis and interpretation. J.J.J.v.d.H. and F.H. supervised this work.

## Competing interests

J.J.J.v.d.H. is a member of the Scientific Advisory Board of NAICONS Srl., Milano, Italy and consults for Corteva Agrisience, Indianapolis, IN, USA. R.S. is a co-founder of mzio GmbH, Bremen, Germany. All other authors declare to have no competing interests.
