## [Transparent Peer Review file · Nature Communications]

Cross ionization mode chemical similarity prediction between tandem mass spectra in metabolomics

Corresponding Author: Mr Niek de Jonge

Version 0:

Reviewer comments:

Reviewer #1

(Remarks to the Author)

The authors introduce an updated version of their tool MS2DeepScore, where the main difference between the previous version is the ability to leverage the information from both positive and negative ionization modes. While tools like MS2DeepScore are a good addition to the field, and new versions of the tool are welcome, this article does not report a groundbreaking improvement. In addition, the results of the assessment of the superiority of the new method are not convincing. These observations are sustained on the following facts:

- Only one urine sample matrix has been used: it is difficult to assess the improved performance with only one dataset, particularly urine.
- In addition, these urine samples are fractions of the same sample collected with semi-preparative chromatography. In that sense, these samples do not represent the type of samples that scientists use and do not showcase the new method's performance for routine metabolomics analysis.
- The comparison between the single ionization mode/ and this new cross/dual mode reports that the difference in performance is not that significant or non-existent when considering only the positive mode.
- There is no major positive/negative molecule overlap, meaning that the same molecules are not usually observed in both modes. Therefore, the utility of this new method precludes its application to a wide range of molecules. This could explain why the only visual example that the authors report in figure 2 is very poor: the positive mode spectrum essentially consists of the protonated peak in addition to a few small fragments. In real cases, it would be difficult to discern whether these small peaks are actual fragments or noise. I also wonder why this spectrum was not flagged as 'low information content' by the new algorithm.

Overall, I believe that the utility of their already published tool MS2DeepScore is undoubtful, and the fact that authors keep maintaining, updating, and integrating it into the Mzmine environment is also very commendable. However, the results of this paper do not describe a significant improvement. I would have endorsed the publication of this manuscript to Nat. Comms. if the authors had demonstrated that this method is distinctly superior to the existing alternatives, also by confirming the identity of unknowns discovered through this approach.

I also have the following concerns:

- The core of the results, paragraph 186-201, reports the main improvements in a very short and general manner. More detailed analysis and discussion are needed, e.g: boxplots comparing the Tanimoto similarity between single/dual mode; more specific examples like that in fig 2, etc.
- How many molecules are shared across modes? This is not discussed in the manuscript. If this overlap is low, the utility of the new method, although welcomed, is limited.
- Lines 133-134: "A model without cross-ionization mode predictions would not have been able to link positive and negative

ionization mode spectra in this cluster together". Could the authors prove this statement? Report the single/dual mode comparison, discuss the improvement in terms of number of clusters, annotation improvement, etc.

- The manuscript is sometimes hard to follow and should be revised for clarity. For example, it is hard to understand what Figure 3 is depicting: e.g, the term 'Percentage of spectra' is shown in the figure, but it is not mentioned anywhere in the manuscript.

- Also regarding this spectra removal, how is this MSE threshold intuitively set by the user? What is the sample size after removal?

Minor:

- Line 173: MSE is not defined.

(Remarks on code availability)

Reviewer #2

(Remarks to the Author)

Authors present a novel method that allows prediction of chemical similarity between spectra generated across the same or different ionization modes. Advances of this software compared to the previous version includes faster runtime, improvements in balancing spectra selection and technical improvements (e.g., hyperparameterization, moving to PyTorch, additional of training pipeline, etc.). The software is freely available, and the method provides a practical solution to a major challenge in the metabolomics field, that of providing structural annotations to metabolite features detected in mass-spectrometry based experiments. Generally, this work is highly valuable for the metabolomics community and authors have conducted a commendable amount of work. However, the organization of the manuscript and lack of clear references to relevant sections of the manuscript (e.g., specific figure numbers) make the manuscript a bit difficult to follow at times. The specific comments will hopefully provide some guidance in clarifying some key points.

Major comments:

- Generally, the results section is quite brief with little explanation as to the reasoning for the experiments and the results. Some of the results also seem incomplete. Examples include:
 - o It would be useful to have a workflow/summary figure that explains the experimental design, utility/use case and how one can build their own model.
 - o The explanation for the training, validation and test sets is explained in the Methods yet can be confusing when reading through the figures. It would be useful to clarify in the Results section when the train, validation or test sets are used and for what purpose for clarity and easier navigation of the figures (main and supplementary). This could also be part of a workflow figure.
 - o In the Sampling algorithm section of the results, there is little explanation regarding the imbalance in the Tanimoto score distribution which would help explain the need for balancing of the training data in the first place for building the model. While this is explained in the Methods and Discussion, it's difficult to understand the results without that context.
 - o Only one dataset has been explored for the case studies section with only one example for ESI+ and ESI- spectra corresponding to a single molecule. Showing more examples and explaining how these spectra corresponding to a single molecule are practically found would be very helpful. The first sentence of the Discussion states that "MS2DeepScore is able to make reliable predictions between mass spectra measured under different conditions, even if hardly any of the fragments overlap." Yet it's unclear how often the algorithm can group together spectra on the same molecule when there is little overlap.
 - o Lines 248-250 describes an important result in the discussion regarding an imbalance in ESI+ and ESI- data. However, the discussion nor the results are very quantitative in the description: what is the average difference MSE between models trained only on ESI- spectra vs. dual model spectra? From Supp Figure 3.3, the differences are in the order of 0.02. Are those meaningful?
- While authors demonstrate clearly the value of MS2DeepScore in predicting chemical similarity across spectra obtained from ESI+ and ESI-, there are still spectra that show high true chemical similarity and lower predicted chemical similarity and vice versa. The number of such spectra is impossible to determine since authors do not provide number of spectra per bin. In Figure 1, it is also very difficult to tell the difference between a density of 0 and a density of 0.05. Have authors explored what types of molecules show substantial differences between true and predicted chemical similarity?
- Figure 1: Since the density represents the percentage of feature pairs that have the same range of predicted and true chemical similarity, a max of 20% density in equal bins between predicted and true chemical similarity seems small. Could authors clarify how and whether this is due to imbalanced molecule sampling as explained in the Supplementary Materials? Further, along the $y=x$ axis, the density increases with increasing chemical similarity. Could authors explain this trend?
- UMAP from Figure 2b:
 - o The resolution is not high enough to see whether both spectra corresponding to cholic acid are adjoining points. Could authors clarify? From a usability standpoint, are users expected to select pairs of nearby molecules and inspect their spectra to determine whether they represent the same molecule?
 - o Could authors clarify how the UMAP representation is complementary to the molecular network visualization? More information on how to navigate each type of graph would be helpful.
 - o At the end of the Figure 2 legend, authors note that the zoomed in portion shows the area with the most spectra, but this

does not seem to be the case. Which is the zoomed in portion? The graph in Figure 2b on the left is already a zoomed in graph compared to the online html provided in the Supplementary Materials.

- Lines 190-196:

- o This text is difficult to follow. For example, “The model trained on only positive ionization mode spectra performs comparable at predicting similarity...”. Is referring to comparing Figure 1b vs. Supp Figure 3.1b or is this referring to Supp Figure 3.3? Explicitly stating so in the text would be tremendously helpful.

- o Further, it’s unclear why authors state that Supp Figure 3.2a and Figure 1a seem to be different (or Supp Figure 3), as stated in the sentence starting with “But the model trained on only negative ionization mode spectra, performs differently at predicting similarity between ...”. Could authors clarify? What does different mean? In the discussion, authors do mention in the discussion the imbalance in the ionization modes represented in the training data yet do not provide any quantification (e.g., ratio of ESI+ to ESI- data?).

- o Finally, what do authors mean by Tanimoto bins having a “higher loss” in one model compared to another?

- Part of “Uncertainty Evaluation” Figure 3B: Can authors clarify what is on the y-axis in Figure 3C? Presumably, it represents the true MSE and the percentage of spectra represents the percentage after removing the top X% spectra based on their predicted MSE.

- Lines 306-307, authors state that spectra that have little similarity yet low quality tend to have very similar embeddings. How often does this occur? While Figure 3b shows a strong correlation between true and predicted MSE, there are still many spectra that do not show strong agreement between the 2. In fact, it seems that predicted MSE values are generally a bit lower than the true MSE (many more points are falling between the x~y line). Have authors explored whether there the differences in true and predicted MSE are associated with the size or class of metabolites representing the spectra? The ones that have a high MSE, are they true for false positives?

- While the example use case shown by authors is very interesting, it does not provide clarity on the utility of the tool. As a user, how does one find examples such as cholic acid? How does one best explore the data? UMAP or molecular networks? While the tool itself is theoretically useful, the practicality on how to use the tool is lacking.

- In the methods it is mentioned the improvements to MS2DeepScore 2.0 has “resulted in a substantial speed-up in the training of models”. There is no comparison metrics provided to show the improved speed.

- Supplementary Figure 1.1

- o What is the rationale for training on positive ionization mode data but using both ionization modes from the validation set?

- o The legend for this figure is unclear. For example (a) is referring to 2 models (authors write “both models”), but it is unclear which 2 models they are referring to.

- Supplementary Figure 1.2 and 1.3: Can authors clarify what ‘dense layer dimensions’ mean? Additionally, why are they separate figures? The legend is exactly the same?

Minor comments:

- Lines 251-258: authors describe the inequality in the number of pairs evaluated per Tanimoto bin. The total number of pairs within a Tanimoto bin are not shown however in Figure 1 or other figures. Those are important to show and better interpret the density metrics.

- While supplementary figures are referenced by section (Supp Fig. 1, Supp Fig. 2, etc.), many are not referenced individually (Supp. Fig. 1.1, etc.) and are associated with distinct results and interpretations.

- Figure 1 and 3B: Consider having the same scale for both x- and y-axes, making it easier to see the highest density along the y=x line.

- Figure 3B: please clarify in the legend how many spectra in total are evaluated. Consider changing all figures with densities (e.g., Fig. 1 and supp figures) to number of spectra or showing densities in main figures and number of spectra in supplementary figures.

- There are a few inconsistencies with labeling of figures:

- o Line 115: Figure 4C should be Figure 1c

- o Line 360: Figure 1 should be Figure 4

- o Line 526: Supplemental Table 7.1 should be Supplemental Table 7.2

- Supp Materials, section “Balanced molecule sampling”. In first paragraph, authors refer to Figure 1A yet Figure 1A does not directly show the inequality of sampling.

- Legend of Figure 2: it would be useful to clarify explicitly that the MS2DeepScore scores are used as edges.

- Lines 462-464: It seems that each embedding evaluated for the training of the model was that of the training data only? The wording is a bit unclear.

- ‘mzmine’ should be written as ‘MZmine’

- Supplementary Figure 5.3: Is this for the test set?

- Supplementary Figure 5.1 and 5.2: What is the ‘n’ for each boxplot? Violin plot might show distributions a bit better?

- Supplementary Figures 6.1 and 6.2: title of the figure needs to be changed to say subset of test set.

- ‘Uncertainty evaluation’

- o The authors have three panels in Figure 3 but only discuss Figure 3c.

- o Can the authors clarify when the authors say, ‘We can improve prediction reliability by filtering out spectra with a high predicted MSE’. Does this mean that for each molecule pair, spectra with a high MSE are removed?

- o It’s unclear what the cleaning process for different analyses: for example, In Supplementary Figure 4.2a, the lowest number of fragments in the boxplot is presented as 1 but in the methods section under “Case Studies” it is mentioned that only MS2 spectra with more than four fragments are kept. Are these different data?

(Remarks on code availability)

(Remarks to the Author)

(Remarks on code availability)

Version 1:

Reviewer comments:

Reviewer #1

(Remarks to the Author)

1.2: The authors reply to my previous point, that "we do not think that the different method of chromatography [...] represent invalid examples. The advantage of using the fractions is that the metabolites had been partly separated and, in comparison to the raw urine, they can be more easily detected and therefore fragmented thanks to the reduced ion suppression."

And this is precisely the problem. If you use data generated to contain richer compound information ("easily detected", meaning that more compounds and with better MS/MS data will be recorded), you will likely find more examples of the utility of the tool. But if this is not the kind of data typically used in metabolomics, then the demonstration of the utility of the tool is biased, and might suggest that the utility of the tool is limited. I appreciate, however, that the authors have included other datasets.

1.3. My point was specifically focused on comparing single ionization mode vs the cross/dual mode presented in the study. From the response, the authors understood my concern, but instead argue that this is not what they were trying to demonstrate: they wanted to show the improvement over the previously published version. From the figure, the performance improvement is subtle. In any case, I disagree with this response. The tool's ultimate goal is to make use of the potentially richer information that can be extracted when jointly using pos and neg modes, compared to single (either pos or neg) mode. And the authors, with their own results, showed that there is virtually no difference. Therefore, the utility of the method is questionable, considering (see previous point) that the authors even used data intended to improve this scenario.

1.4 While I appreciate the addition of Fig 10.2, for the case of Fig 10.1, which is included in response to my concern, the figure shows the same example multiple times, therefore, not addressing my concern. Now, the authors include a new Fig 2 where they show a "similarity correlation" plot to demonstrate the tool's performance. The standard deviation is very high and this analysis only shows a general performance trend but does not analyze the performance in depth. I suggest including a qualitative analysis where top-k results are shown, e.g., in how many cases the most similar molecule is predicted within top-1, top-3, top-N predictions; and also include an additional quantitative analysis to show the relative true-predicted similarity error. These analyses should be triplicated to compare pos vs neg vs the dual mode to show the advantages of this dual mode. I am referring to the actual application of the tool, not whether the tool has been trained with only one mode or dual mode data.

Overall, my concerns still stand, and they are aligned or very similar to those of Reviewer #2, a fact that reinforces their validity. One of the other reviewers' concerns is about the shared molecules across modes. The authors' response included an analysis of a database instead of using real data samples. This also shows that the authors can not demonstrate the advantage of combining both modes. Taking the results, my points and the authors' response together, it seems that the tool is limited by the poor information that the suggested dual mode can contribute to the identification of molecules. While I appreciate the study within this manuscript, I can not endorse this manuscript for publication in Nat. Comms for the same reasons stated in my previous report.

(Remarks on code availability)

Reviewer #2

(Remarks to the Author)

The authors have greatly improved the clarity of the results, methods and manuscript in general. Please ensure that the answers to the queries provided are included in the results and discussion as appropriate. Several minor specific points to note:

- Regarding point 2.2.1: the replacement of the heatmaps by the violin plots is very helpful. However, because the scales are different (necessarily so to create the violins), it's difficult to visualize where the $y=x$ line falls. Can this be somehow displayed for easier reading of the plots?
- Regarding point 2.2.3: while authors reference Supp Figures 7.1 and 7.2 in the text and provide an explanation in the rebuttal, they do not explain in the text that certain adducts have higher MSE. It would be useful to clarify this in the text. The same comment holds for Supp Figure 6.1.

- Regarding 2.4.1: Authors clarify in the rebuttal that users need to manually differentiate between exact matches and closely related molecules and could use RT and mass differences to help with this task. This could be clarified in the text, as well as potential future automation of this.

(Remarks on code availability)

Reviewer #3

(Remarks to the Author)

(Remarks on code availability)

Version 2:

Reviewer comments:

Reviewer #1

(Remarks to the Author)

I have no further comments.

(Remarks on code availability)

REVIEWER COMMENTS

Reviewer #1 (Remarks to the Author):

The authors introduce an updated version of their tool MS2DeepScore, where the main difference between the previous version is the ability to leverage the information from both positive and negative ionization modes. While tools like MS2DeepScore are a good addition to the field, and new versions of the tool are welcome, this article does not report a groundbreaking improvement. In addition, the results of the assessment of the superiority of the new method are not convincing. These observations are sustained on the following facts:

1.1 Only one urine sample matrix has been used: it is difficult to assess the improved performance with only one dataset, particularly urine.

We agree that one case study may bias the performance assessment of MSDeepScore's applicability. Therefore, we have now added two additional case studies. One case study a plant sample (*Rumex sanguineus*) and another study that measured human blood plasma samples. We have updated Figure 3 and replaced the Urine case study UMAP with a UMAP visualization of the *Rumex sanguineus* case study. Supplementary Figure 9 shows a molecular network for the newly added blood plasma case study, with manually curated annotations. The newly added case studies also have multiple cross-ionization mode clusters, with correctly predicted connections. This highlights that MS2DeepScore works well over a diverse set of samples.

Still, it is important to note that the case studies are not intended to judge general performance, since performance can vary highly from case study to case study (see for instance, previous work on MS2Query¹). The goal of the case studies is to better illustrate the practical impact on real datasets. To judge the general performance, the benchmarking on the test set (Figure 2) gives a more accurate representation of performance. Firstly because, the test set contains a random selection of metabolites, which covers a more diverse set of metabolites. Secondly the test sets are larger (32052 spectra) and for the test set we know the ground truth for all spectra.

¹ de Jonge, N.F. et al. MS2Query: reliable and scalable MS2 mass spectra-based analogue search. *Nature Communications* **14**, 1752 (2023).

a) Molecular network urine case study

b) UMAP embedding representation *Rumex sanguineus* (plant) case study

Figure 3: Visual sample representations created with MS2DeepScore cross-ionization-mode model predictions. By predicting chemical similarity between both the positive and negative ionization mode spectra, spectra of both ionization modes can be visualized together. a) A molecular network of the urine case study created by using MS2DeepScore similarity scores. An edge is created for an MS2DeepScore larger than 0.85. We highlight a few examples where MS2DeepScore was able to predict close chemical similarity between positive and negative ionization modes. Spectrum mirror plots of all cross-ionization mode pairs visualized in the molecular network can be found in Supplementary Figure 10.1. Spectrum mirror plots for exact matches across ionization modes are available in Supplementary Figure 10.2. An interactive version of the molecular network can be loaded in Cytoscape, the data is available via the Data Availability section. b) UMAP representation of the MS2DeepScore 2.0 embeddings of the *Rumex sanguineus* case study. Each dot represents a spectrum and closely positioned spectra have high predicted chemical similarity. The molecular structure for multiple annotated mass spectra are visualized as examples. Two spectra are highlighted which both correspond to the same molecule, but were recorded in positive and negative ionization modes. MS2DeepScore 2.0 correctly predicted very similar embeddings, while the fragments do not overlap. An interactive version of the full UMAP plot is available as an HTML file, see Data Availability section.

Supplementary Figure 9: Molecular network created with MS2DeepScore cross-ionization-mode model on human blood plasma case study. By predicting chemical similarity between both the positive and negative ionization mode spectra, spectra of both ionization modes can be visualized together. An edge is created for an MS2DeepScore larger than 0.85. We highlight a few examples where MS2DeepScore was able to predict close chemical similarity between positive and negative ionization modes. The annotations were added by manual annotations.

1.2 In addition, these urine samples are fractions of the same sample collected with semi-preparative chromatography. In that sense, these samples do not represent the type of samples that scientists use and do not showcase the new method's performance for routine metabolomics analysis.

In our revised manuscript, we have now added an additional case study on a plant dataset, which was generated in a more traditional manner. This newly added case study shows similar results as the original case study.

In addition, we do not think that the different method of chromatography in the Urine case study (and the human blood case study) represent invalid examples. Fractions were only generated with semi-prep chromatography but analysed by UHPLC-MS and MS/MS. The advantage of using the fractions is that the metabolites had been partly separated and, in comparison to the raw urine, they can be more easily detected and therefore fragmented thanks to the reduced ion suppression. This allows it to detect a wider range of metabolites.

1.3 The comparison between the single ionization mode/ and this new cross/dual mode reports that the difference in performance is not that significant or non-existent when considering only the positive mode.

The training on both ionization modes at the same time was not intended to improve the within-ion-mode performance. Instead, it was intended to enable cross-ionization mode predictions, which is an entirely novel functionality not possible with any prior technique. However, many other improvements were made in this work that improve the within-ion-mode performance.

This can be seen in the newly added direct comparison to the original MS2DeepScore model (version 0.2.0) in **Supplementary Section 3**. The displayed improvements are a result of the addition of the precursor m/z as metadata, the optimized sampling algorithm, and the hyperparameter optimization. We also added a section in the results of the main text describing this comparison explicitly.

Supplementary Section 3. Comparison to MS2DeepScore 0.2.0 model

Supplementary Figure 3.1: Comparison between current MS2DeepScore version (2.5.4) and the original MS2DeepScore version (0.2.0). The MS2DeepScore model 0.2.0 is the version used to train the model with the same architecture as the original MS2DeepScore paper. This model is retrained on the same training data and benchmarked on the same test set to enable comparison to the new MS2DeepScore version (2.5.4). For the 0.2.0 architecture, two models are trained, one on positive ionization mode and one on negative ionization mode spectra. Predictions are made between all test spectra, followed by taking the average per unique molecule pair. The violin plots show the kernel density estimation (KDE) of the predicted values, the black lines represent the median. The bar plot on the top shows the log-scaled count of the number of unique molecule pairs in each bin with the corresponding chemical similarity. The metric used for chemical similarity prediction is the Tanimoto score between Daylight fingerprints. a) Predictions between pairs of negative ionization mode spectra. b) Predictions between pairs of positive ionization mode spectra.

Supplementary Figure 3.2: MSE per Tanimoto bin for current MS2DeepScore version (2.5.4) versus original MS2DeepScore version (0.2.0). Compound pairs are sampled from the test set per Tanimoto bin. The average MSE per compound pair is calculated followed by calculating the average over all compound pairs in the Tanimoto bin. a) Predictions between pairs of positive ionization mode spectra. b) Predictions between pairs of negative ionization mode spectra.

Comparison to the original MS2DeepScore model

This paper introduces multiple advancements to the original MS2DeepScore paper, including adding metadata and the optimized sampling algorithm. To compare the performance, the original MS2DeepScore model (version 0.2.0) was retrained and tested on the same training and test set used here. Supplementary Figure 3.1 shows side by side violin plot, directly comparing the performance of the original MS2DeepScore model to the new MS2DeepScore model. Supplementary Figure 3.2 shows the average MSE per Tanimoto bin, showing that the performance is improved by the new training approach introduced here.

1.4: There is no major positive/negative molecule overlap, meaning that the same molecules are not usually observed in both modes. Therefore, the utility of this new method precludes its application to a wide range of molecules. This could explain why the only visual example that the authors report in figure 2 is very poor: the positive mode spectrum essentially consists of the protonated peak in addition to a few small fragments. In real cases, it would be difficult to discern whether these small peaks are actual fragments or noise. I also wonder why this spectrum was not flagged as 'low information content' by the new algorithm.

With regard to not observing the same molecule in both ion modes: One of the strengths of this model is that it can predict chemical similarity across ionization modes even if they do not have identical structures. Many examples in the molecular network in Figure 2 support this by showing predicted high similarity between chemically closely related molecules. There is therefore no need to have both ion mode spectra for a single molecule to make a prediction. Whilst not the main goal of MS2DeepScore, we note that there are multiple cases of identical molecules recorded in positive and negative ionization mode that have a high predicted chemical similarity by MS2DeepScore. We have added spectral comparisons for these cases in Supplementary Figure 10.2.

Regarding the low intensity of the fragments in the example spectra in Figure 3: The high intensity of the complete ion, indeed makes it hard to judge the quality of the negative ionisation mode spectrum. Below, we added a zoomed-in version of this figure to show that this is not just noise. We picked this example spectrum pair, since it highlights that our model is able to correctly predict high chemical similarity even if the spectra are visually very distinct. We have now replaced the UMAP example in Figure 3b with the results of the *Rumex sanguineus* case study, therefore highlighting a different example pair, see reply to comment 1.1.

Instead of just showing a single spectrum comparison, we have now added more examples of cross ionization mode spectrum pairs, with high predicted similarity in Supplementary Figure 10.1 and 10.2. Supplementary Figure 10.1 shows the spectra pairs for all cross-ionization mode predictions annotated in the molecular network in Figure 3a, and in addition, we added 8 examples of identical matches across ionization modes.

The following is added to the Supplementary information:

Supplementary Figure 10.1: Spectrum comparisons of positive and negative ionization mode spectra for which MS2DeepScore predicts high chemical similarity. The spectra are from the urine case study. Positive ionization mode spectra are blue and negative ionization mode spectra are orange. All annotated spectra connected across ionization modes in the molecular network in Figure 3 are visualized.

Supplementary Figure 10.2: Spectrum comparisons of positive and negative ionization mode spectra from the urine case study. Structures are putatively annotated through MS2Query predictions and all have a MS2Query score of at least 0.8 and a mass difference < 0.1 for the precursor m/z . Spectrum pairs are selected for which a positive and negative ionization mode spectrum was detected, with identical annotation. Positive ionization mode spectra are blue and negative ionization mode spectra are orange. The cosine score, the modified cosine score and the MS2DeepScore prediction are given between each spectrum pair.

Overall, I believe that the utility of their already published tool MS2DeepScore is undoubtful, and the fact that authors keep maintaining, updating, and integrating it into the Mzmine environment is also very commendable. However, the results of this paper do not describe a significant improvement. I would have endorsed the publication of this manuscript to Nat. Comms. if the authors had demonstrated that this method is distinctly superior to the existing alternatives, also by confirming the identity of unknowns discovered through this approach.

Thank you for all useful comments that have helped us to further clarify the novelty of our new model by adding additional case studies and improving overall clarity of text and figures. To us, the ability to reliably predict chemical similarity across ionization mode is an exciting novel direction for the field that can have direct practical applications to many users.

I also have the following concerns:

1.5 The core of the results, paragraph 186-201, reports the main improvements in a very short and general manner. More detailed analysis and discussion are needed, e.g: boxplots comparing the Tanimoto similarity between single/dual mode; more specific examples like that in fig 2, etc.

We realize this section was not written clearly. We have extended and rewritten this section to clarify the goals of this comparison. We updated Supplementary Figures 4.1-4.3 to use side-by-side violin plots to make a direct comparison with the single ionization mode models. In addition, we also added a direct comparison to the model architecture used in the original MS2DeepScore paper, showing that the new architecture and training methods lead to improved accuracies for within ionization mode comparisons, besides the novel functionality of cross-ionization mode predictions.

In addition, we would like to note that the core of the results to us are not these comparisons, but instead the newly added cross-ionization mode prediction capabilities. We have updated other sections of the results as well, to make this clearer.

Rewritten in the main text:

Comparison to the original MS2DeepScore model

This paper introduces multiple advancements to the original MS2DeepScore paper, including adding metadata and the optimized sampling algorithm. To compare the performance, the original MS2DeepScore model (version 0.2.0) was retrained and tested on the same training and test set used here. Supplementary Figure 3.1 shows side by side violin plot, directly comparing the performance of the original MS2DeepScore model to the new MS2DeepScore model. Supplementary Figure 3.2 shows the average MSE per Tanimoto bin, showing that the performance is improved by the new training approach introduced here.

Comparison with single ionization mode models

MS2DeepScore 2.0 has the novel capability to predict chemical similarity across ionization modes. However, it is important that the within-ionization-mode prediction accuracy is not affected too much by training on both ionization modes at the same time. The comparison to the original MS2DeepScore model in Supplementary Figure 3.1 and 3.2 already shows an improved performance for the new dual ionization mode model. However, this comparison also includes the here introduced improvements to the model architecture and the inclusion of precursor m/z as input to the model. To only test the effect of training a model on both ionization modes at the same time, two additional models were trained on a single ionization mode using the same hyperparameters as the dual ion mode model. The results are summarized in Supplementary Figure 4.1-4.3. Supplementary Figure 4.3 compares the MSE for the within-ionization-mode predictions for a model trained on both ionization modes and a model trained on only a single ionization mode.

Models trained on only positive ionization mode spectra or on spectra of both ionization modes show comparable within-ionization-mode performance. The model trained on only negative ionization

mode spectra, however, results in lower losses for lower Tanimoto bins, but a higher loss in the 0.9-1.0 Tanimoto bin when compared to the dual-ionization-mode model. Supplementary Figure 4.2 shows a side by side violin plot, enabling direct comparisons between the distributions of predicted scores.

Updated and added to supplementary information: Supplementary Section 3. Comparison to MS2DeepScore 0.2.0 model

Supplementary Figure 3.1: Comparison between current MS2DeepScore version (2.5.4) and the original MS2DeepScore version (0.2.0). The MS2DeepScore model 0.2.0 is the version used to train the model with the same architecture as the original MS2DeepScore paper. This model is retrained on the same training data and benchmarked on the same test set to enable comparison to the new MS2DeepScore version (2.5.4). For the 0.2.0 architecture, two models are trained, one on positive ionization mode and one on negative ionization mode spectra. Predictions are made between all test spectra, followed by taking the average per unique molecule pair. The violin plots show the kernel density estimation (KDE) of the predicted values, the black lines represent the median. The bar plot on the top shows the log-scaled count of the number of unique molecule pairs in each bin with the corresponding chemical similarity. The metric used for chemical similarity prediction is the Tanimoto score between Daylight fingerprints. a) Predictions between pairs of negative ionization mode spectra. b) Predictions between pairs of positive ionization mode spectra.

Supplementary Figure 3.2: MSE per Tanimoto bin for current MS2DeepScore version (2.5.4) versus original MS2DeepScore version (0.2.0). Compound pairs are sampled from the test set per Tanimoto bin. The average MSE per compound pair is calculated followed by calculating the average over all compound pairs in the Tanimoto bin. a) Predictions between pairs of positive ionization mode spectra. b) Predictions between pairs of negative ionization mode spectra.

Supplementary Section 4: Comparison to model trained on single ionization mode

Supplementary Figure 4.1: Comparison between cross-ionization model and model trained only on positive ionization mode spectra. The model trained on only positive ionization mode spectra used the same architecture, but only trained on the positive ionization mode spectra. Predictions are made between all test spectra, followed by taking the average per unique molecule pair. The violin plots show the kernel density estimation (KDE) of the predicted values, the black lines represent the median. The bar plot on the top shows the log-scaled count of the number of unique molecule pairs in each bin with the corresponding chemical similarity. The metric used for chemical similarity prediction is the Tanimoto score between Daylight fingerprints. a) Predictions between pairs of positive ionization mode spectra. b) Predictions between pairs of positive and negative ionization mode spectra. c) Predictions between pairs of negative ionization mode spectra.

Supplementary Figure 4.2: Comparison between cross-ionization model and model trained only on negative ionization mode spectra. The model trained on only negative ionization mode spectra used the same architecture, but only trained on the negative ionization mode spectra. Predictions are made between all test spectra, followed by taking the average per unique molecule pair. The violin plots show the kernel density estimation (KDE) of the predicted values, the black lines represent the median. The bar plot on the top shows the log-scaled count of the number of unique molecule pairs in each bin with the corresponding chemical similarity. The metric used for chemical similarity prediction is the Tanimoto score between Daylight fingerprints. a) Predictions between pairs of positive ionization mode spectra. b) Predictions between pairs of positive and negative ionization mode spectra. c) Predictions between pairs of negative ionization mode spectra.

Supplementary Figure 4.3: MSE per Tanimoto bin for single ionization mode model compared to dual ionization mode model. Compound pairs are sampled from the test set per Tanimoto bin. The average MSE per compound pair is calculated followed by calculating the average over all compound pairs in the Tanimoto bin. a) Predictions between pairs of negative ionization mode spectra. The model trained on single ionization mode spectra performs better than the model trained on both ionization modes. b) Predictions between pairs of positive ionization mode spectra.

1.6 How many molecules are shared across modes? This is not discussed in the manuscript. If this overlap is low, the utility of the new method, although welcomed, is limited.

In the reply to comment 1.4 we showed additional examples of exact matches within the Urine case studies. However, in the case studies, we do not have annotations for most of the data which makes it challenging to get exact numbers on how frequently this occurs. To address your question in more detail, we therefore looked at the public data available. The figure below shows how frequent a molecule was recorded in both positive and negative ionization mode.

In addition, we would like to highlight that MS2DeepScore does a chemical similarity prediction which also works if two molecules are not identical. See, for instance, the Caffeine-related cluster in Figure 3. Here, the connections are not between identical molecules, but between closely related molecules across ion modes. This means that the extent to which our new model can find relevant matches extends the exact overlap of library spectra.

1.7 Lines 133-134: "A model without cross-ionization mode predictions would not have been able to link positive and negative ionization mode spectra in this cluster together". Could the authors prove this statement? Report the single/dual mode comparison, discuss the improvement in terms of number of clusters, annotation improvement, etc. There are no other existing methods that are designed or intended to predict cross-ionization mode similarity. We wanted communicate that the predicted high chemical similarity across ionization mode gives novel information, which other existing methods don't. Since we agree that this sentence is not very clear, we have rewritten the section to: "For example the left cluster contains caffeine-related molecules, all part of known caffeine metabolism pathways²⁴. This shows that MS2DeepScore was able to highlight cross ionization mode connections in the molecular network that correspond to real metabolic pathways."

In addition we also added comparison with already existing methods on the case studies. In Supplementary Figure 10.1 and 10.2, spectral comparisons of the urine case study are visualized. For each spectrum pair we added the cosine score and modified cosine score. For the caffeine cluster, the modified cosine score is never above 0.45, while a common threshold is 0.7 for molecular networking. For another cross-ionization mode connection the modified cosine score is 0.82 (left bottom Supplementary Figure 7.1), yet this is due to a badly fragmented negative ionization mode spectrum, mainly consisting of the precursor ion, and should therefore rather be regarded as an outlier.

1.8 The manuscript is sometimes hard to follow and should be revised for clarity. For example, it is hard to understand what Figure 3 is depicting: e.g, the term 'Percentage of spectra' is shown in the figure, but it is not mentioned anywhere in the manuscript. Thank you for noting. We have improved Figure 3 and the Figure caption. In addition, we have made many improvements throughout the manuscript to improve clarity. For instance, by replacing the heatmaps (e.g. Figure 2), with violin plots, which we expect to be easier to interpret.

The updated version of Figure 3:

c) Removing spectra with high predicted error

Figure 4: Results Embedding Evaluator. The embedding evaluator predicts if MS2DeepScore can make reliable chemical similarity predictions for an input spectrum. This embedding evaluator is trained to predict the mean squared error (MSE) for a spectrum. a) Architecture for the embedding evaluator. b) The predicted MSE is plotted against the true MSE for each spectrum in the test set. The test set contains 32052 spectra. c) Different thresholds are used to remove spectra from the test set with high predicted MSE. This results in 5 subsets of spectra with respectively 20, 40, 60, 80 and 100 % of the test spectra retained. The plot shows the real MSE per bin for predicting chemical similarity for the selected subsets of the test set. This shows that the accuracy of MS2DeepScore improves if the Embedding Evaluator is used to remove spectra with high predicted MSE. The MSE is calculated by taking the average MSE between all spectra of two molecules, followed by taking the average per Tanimoto bin.

1.9 Also regarding this spectra removal, how is this MSE threshold intuitively set by the user? What is the sample size after removal?

We agree that setting the right thresholds can be challenging for (new) users. We have now added the actual thresholds used next to the percentages for the test set to Figure 4. How many spectra are removed for user-specific datasets, will depend on the spectrum quality and similarity to the training spectra. By adding these settings, we believe that this will make it easier for users to pick suitable settings for their use case.

Minor:

1.10 Line 173: MSE is not defined.

Thank you for noting, we have changed MSE to mean squared error (MSE) at the first occurrence in the text.

Reviewer #2 (Remarks to the Author):

Authors present a novel method that allows prediction of chemical similarity between spectra generated across the same or different ionization modes. Advances of this software compared to the previous version includes faster runtime, improvements in balancing spectra selection and technical improvements (e.g., hyperparameterization, moving to PyTorch, additional of training pipeline, etc.). The software is freely available, and the method provides a practical solution to a major challenge in the metabolomics field, that of providing structural annotations to metabolite features detected in mass-spectrometry based experiments. Generally, this work is highly valuable for the metabolomics community and authors have conducted a commendable amount of work. However, the organization of the manuscript and lack of clear references to relevant sections of the manuscript (e.g., specific figure numbers) make the manuscript a bit difficult to follow at times. The specific comments will hopefully provide some guidance in clarifying some key points.

Thank you for your extensive, and very helpful, feedback and suggestions. We believe that by addressing your comments, the paper has become better organized and easier for readers to follow and understand.

Major comments:

2.1 Generally, the results section is quite brief with little explanation as to the reasoning for the experiments and the results. Some of the results also seem incomplete.

Examples include:

2.1.1 It would be useful to have a workflow/summary figure that explains the experimental design, utility/use case and how one can build their own model.

That is a great suggestion. We have added a workflow figure as Figure 1. We merged Figure 4 into this figure.

Regarding the building of your own model, we provide a tutorial in the readme on github. We have now added a sentence in the Code availability section that guides readers that want to train their own model.

Figure 1: a) Training MS2DeepScore. MS2DeepScore is trained on annotated mass spectra from public libraries. The model is trained to predict chemical similarity between pairs of mass spectra. By training the model on both ionization modes at the same time, the model is trained to predict chemical similarity within and across ionization modes. New models can be trained with additional in-house annotated mass spectra to further improve model accuracies. A Siamese neural network architecture is used. The input layer comprises scan metadata and fragment data after binning the m/z axis and applying square root transformation to the signal intensities. Numerical data, e.g., precursor m/z or collision energy, is transformed to values closer to 1, to have input in a similar order of magnitudes, to optimize training. Textual inputs, like ionization mode or instrument type, are one-hot encoded. A single dense layer converts the input to a numerical vector (embedding) of length 500. The model is trained to create embeddings for which the cosine similarity between two embeddings correlates well with chemical similarity (Tanimoto score). **b) Using MS2DeepScore.** MS2DeepScore predicts chemical similarity both within and across ionization modes. By using these predicted similarities to define edges, a unified molecular network can be constructed that connects spectra from both ionization modes.

2.1.2 The explanation for the training, validation and test sets is explained in the Methods yet can be confusing when reading through the figures. It would be useful to clarify in the Results section when the train, validation or test sets are used and for what purpose for clarity and easier navigation of the figures (main and supplementary). This could also be part of a workflow figure.

Each Figure in the main text and supplementary information now clearly mentions if it uses the test set, the validation set, the training set, or the case study data. We also added a short explanation of how the test set is created in the results section.

“To test the performance of MS2DeepScore, a test set of annotated spectra is created, which is not used during the training of the model. The performance of the model is assessed by comparing the predicted chemical similarity by MS2DeepScore with the known true chemical similarity.”

2.1.3 In the Sampling algorithm section of the results, there is little explanation regarding the imbalance in the Tanimoto score distribution which would help explain the need for balancing of the training data in the first place for building the model. While this is explained in the Methods and Discussion, it's difficult to understand the results without that context.

We have added “One of the key challenges in training a model to predict Tanimoto scores is the highly non-uniform distribution of these scores across possible molecule pairs. Low Tanimoto scores are several orders of magnitude more frequent than high Tanimoto scores.” to the results section to clarify this.

2.1.4 Only one dataset has been explored for the case studies section with only one example for ESI+ and ESI- spectra corresponding to a single molecule. Showing more examples and explaining how these spectra corresponding to a single molecule are practically found would be very helpful. The first sentence of the Discussion states that “MS2DeepScore is able to make reliable predictions between mass spectra measured under different conditions, even if hardly any of the fragments overlap.” Yet it's unclear how often the algorithm can group together spectra on the same molecule when there is little overlap.

We have added additional examples of spectrum pairs of the Urine case studies in Supplementary Figure 10.1 and 10.2, which can be found in the response to 1.4.

In practice, it is indeed not straightforward to differentiate a likely analogue across ion modes and a likely exact match. The goal of our model is more focused on linking together mass spectra (across ion modes) that are chemically closely related, but not necessarily exact matches. This makes annotation propagation and clustering possible even if there is no molecule for which ESI+ and ESI- has been recorded. Our model does not make a clear

differentiation between likely chemically very similar and likely an exact match. However, under the same experimental conditions, it is possible to use retention time to link two spectra belonging to the same molecule together. The real novelty and strength of our model is that we can predict chemical similarity across ion modes even if none of the molecules in a sample is recorded in both ionization modes.

2.1.5 Lines 248-250 describes an important result in the discussion regarding an imbalance in ESI+ and ESI- data. However, the discussion nor the results are very quantitative in the description: what is the average difference MSE between models trained only on ESI- spectra vs. dual model spectra? From Supp Figure 3.3, the differences are in the order of 0.02. Are those meaningful? We agree that it is hard to intuitively judge what the impact is of a difference in MSE in practice. To improve the ease of interpretation, we have changed the heatmaps into violin plots directly comparing the model trained on only one ion mode and both ion modes. Supplementary Figure 4.2c shows that these differences are not substantial. In addition we now added a comparison to the original MS2DeepScore model in Supplementary Figure 3.1 and 3.2. These figures show that the improvements made by optimizing the architecture and the training methods have a larger positive impact on model performance.

Although the comparison to the single ionization mode model does not show a very substantial difference, it is still an interesting finding that can guide future work. In future work it will be valuable to try oversampling negative ionization mode spectra, to further improve performance for within ion mode predictions.

Below the new Supplementary Figure 4.2. The other mentioned supplementary Figures can be found in the response to reviewer 1, comment 1.5.

Supplementary Figure 4.2: Comparison between cross-ionization model and model trained only on negative ionization mode spectra. The model trained on only negative ionization mode spectra used the same architecture, but only trained on the negative ionization mode spectra. Predictions are made between all test spectra, followed by taking the average per unique molecule pair. The violin plots show the kernel density estimation (KDE) of the predicted values, the black lines represent the median. The bar plot on the top shows the log-scaled count of the number of unique molecule pairs in each bin with the corresponding chemical similarity. The metric used for chemical similarity prediction is the Tanimoto score between Daylight fingerprints. **a)** Predictions between pairs of positive ionization mode spectra. **b)** Predictions between pairs of positive and negative ionization mode spectra. **c)** Predictions between pairs of negative ionization mode spectra.

2.2.1 While authors demonstrate clearly the value of MS2DeepScore in predicting chemical similarity across spectra obtained from ESI+ and ESI-, there are still spectra that show high true chemical similarity and lower predicted chemical similarity and vice versa. The number of such spectra is impossible to determine since authors do not provide number of spectra per bin.

We agree that this is hard to get from the heatmap. We realized that the heatmap did not clearly depict that the normalization was done per Tanimoto bin, which can lead to confusion for the reader. Therefore, we replaced the heatmap in Figure 1 and all heatmaps in the supplementary information with violin plots. We added a bar plot to this violin plot to show the number of unique molecule pairs in the test set per bin.

Figure 2: Dual-ionization mode MS2DeepScore model predicts chemical similarity between and across ionization modes. A test set of 32052 spectra is used, which were not used to train the model. Predictions are made between all test spectra, followed by taking the average per unique molecule pair. The violin plots show the kernel density estimation (KDE) of the predicted values, the black lines represent the median and the 1st and 99th percentile for each bin. The bar plot on the top shows the log-scaled count of the number of unique molecule pairs in each bin with the corresponding chemical similarity. The metric used for chemical similarity prediction is the Tanimoto score between Daylight fingerprints. a) Predictions between pairs of positive ionization mode spectra. b) Predictions between pairs of positive and negative ionization mode spectra. c) Predictions between pairs of negative ionization mode spectra.

2.2.2 In Figure 1, it is also very difficult to tell the difference between a density of 0 and a density of 0.05.

We have replaced Figure 1 (now Figure 2) with violin plots, which also show the low frequencies more clearly.

2.2.3 Have authors explored what types of molecules show substantial differences between true and predicted chemical similarity?

In Supplementary Figure 7.1 and 7.2 we show an overview of the MSE per adduct type and per chemical class, giving some insights into the general patterns of challenging cases. Between the different compound classes, we did not observe any striking differences (Figure 7.2). Compounds of the class “nucleosides, nucleotides, and analogues” seem to receive slightly better chemical similarity predictions, but we consider this effect rather small. When comparing different adducts, however, we noted that the adducts $[M+Na]^+$ and $[M+K]^+$ have a higher MSE and therefore have a larger difference between true and predicted chemical similarity.

2.3.1 Figure 1: Since the density represents the percentage of feature pairs that have the same range of predicted and true chemical similarity, a max of 20% density in equal bins between predicted and true chemical similarity seems small.

We have replaced figure 1 with a violin plot, which gives more insight into the exact distribution per bin.

The reason the heatmap showed max densities of 20% in equal bins, is that the density mostly depends on the size of the bins that we show. Since we used 50 bins, the accuracy has to be very high to all end up in the same bin (within 0.02 Tanimoto score). This is not the kind of accuracy that we reasonably expect.

By choosing smaller bins, the highest density goes up, since the size of a bin is larger. To illustrate this, see the Figure below, where a bin size of 10 was used. Here, the highest density is 0.5, but some detail is lost.

2.3.2 Could authors clarify how and whether this is due to imbalanced molecule sampling as explained in the Supplementary Materials?

The new sampling algorithm developed here ensures well-balanced molecule sampling over the different bins. So we don't expect that further optimization of the sampling algorithm would improve performance within an ionmode.

However, the pair sampling is not yet perfectly balanced for the different ion modes. In the discussion we discuss this in this section: "In addition, since there are differences in the Tanimoto score distributions for positive and negative ionization mode spectra, this might result in not having an equal number of pairs per Tanimoto bin for each ionization mode. For example, since there were not many high Tanimoto score examples between positive and negative ionization mode spectra during training, it is likely that this explains the observation that almost no predictions above 0.9 were made for cross-ionization mode pairs. In future work, the sampling algorithm could be further optimized to also enforce balanced pair sampling for the different ionization mode pairs."

2.3.3 Further, along the $y=x$ axis, the density increases with increasing chemical similarity. Could authors explain this trend?

These high densities were mostly an artifact of the heatmap visualization method, by replacing the heatmap with the violin plot, this is not an issue anymore.

These higher density spots are due to having less unique pairs in the test set for the high chemical similarity bins. Since the sample size is so small for these higher bins (see figure below for exact distribution), random fluctuations can result in high densities by chance. For low chemical similarities we have many available pairs, leading to a smooth transition between the different predicted bins.

2.4 UMAP from Figure 2b:

2.4.1 The resolution is not high enough to see whether both spectra corresponding to cholic acid are adjoining points. Could authors clarify? From a usability standpoint, are users expected to select pairs of nearby molecules and inspect their spectra to determine whether they represent the same molecule?

Our model predicts chemical similarity and not directly predicts if two spectra are an exact match. In the UMAP if two spectra are positioned closely together, this indicates high predicted similarity. It can therefore indeed be more challenging to differentiate between exact matches and closely related molecules. In the case of an exact match, a user could use additional information such as retention time and mass difference to check if it is likely an exact match. In future work, it would be interesting to try to automate this process, for instance by integrating MS2DeepScore with Ion identity molecular networking across ionization modes.

2.4.2 Could authors clarify how the UMAP representation is complementary to the molecular network visualization? More information on how to navigate each type of graph would be helpful.

We could indeed have clarified this better in the paper. Thank you for noting this. To highlight this more clearly in the manuscript, we added the following sentences to the results section:

“MS2DeepScore also enables alternative visualization methods that overcome some limitations of molecular networking. One limitation of molecular networking is that it does not depict relationships between separate clusters. A second challenge with molecular networking is choosing a hard cut-off for when connecting an edge. If this threshold is set too high, relevant connections are not depicted, while if this threshold is set too low a hairball of connections can form, which makes it challenging to interpret the data. An alternative visualization method is using UMAP to visualize MS2DeepScore embeddings, this overcomes both these limitations of molecular networking. Each datapoint in the UMAP represent one spectrum and spectra located closely together in the UMAP have high predicted chemical similarity.”

2.4.3 At the end of the Figure 2 legend, authors note that the zoomed in portion shows the area with the most spectra, but this does not seem to be the case. Which is the zoomed in portion? The graph in Figure 2b on the left is already a zoomed in graph compared to the online html provided in the Supplementary Materials.

The left figure was indeed already zoomed in. We intended to clarify this with the last sentence. We now realize that this is confusing, since Figure 2b also contains a zoomed-in plot.

We have now replaced the UMAP, with a UMAP of the *Rumex sanguineus* case study. We also removed the zoom, to allow for adding more example structures to the figure.

2.5 Lines 190-196:

2.5.1 This text is difficult to follow. For example, “The model trained on only positive ionization mode spectra performs comparable at predicting similarity...”. Is referring to comparing Figure 1b vs. Supp Figure 3.1b or is this referring to Supp Figure 3.3? Explicitly stating so in the text would be tremendously helpful.

This was indeed written confusingly. We have rewritten the entire section. In addition, we have changed supplementary Figure 3.1 and 3.2 to use comparison violin plots (now 4.1 and 4.2). This makes it possible to directly compare the performance in the same figure, without having to compare to Figure 1. The rewritten section is:

Comparison with single ionization mode models

MS2DeepScore 2.0 has the novel capability to predict chemical similarity across ionization modes. However, it is important that the within-ionization-mode prediction accuracy is not affected too much by training on both ionization modes at the same time. The comparison to the original MS2DeepScore model in Supplementary Figure 3.1 and 3.2 already shows an improved performance for the new dual ionization mode model. However, this comparison also includes the here introduced improvements to the model architecture and the inclusion of precursor m/z as input to the model. To only test the effect of training a model on both ionization modes at the same time, two additional models were trained on a single ionization mode using the same hyperparameters as the dual ion mode model. The results are summarized in Supplementary Figure 4.1-4.3. Supplementary Figure 4.3 compares the MSE for the within-ionization-mode predictions for a model trained on both ionization modes and a model trained on only a single ionization mode.

Models trained on only positive ionization mode spectra or on spectra of both ionization modes show comparable within-ionization-mode performance. The model trained on only negative ionization mode spectra, however, results in lower losses for lower Tanimoto bins, but a higher loss in the 0.9-1.0 Tanimoto bin when compared to the dual-ionization-mode model. Supplementary Figure 4.2 shows a side by side violin plot, enabling direct comparisons between the distributions of predicted scores.

2.5.2 Further, it's unclear why authors state that Supp Figure 3.2a and Figure 1a seem to be different (or Supp Figure 3), as stated in the sentence starting with "But the model trained on only negative ionization mode spectra, performs differently at predicting similarity between". Could authors clarify? What does different mean?

After rereading, we agree to improve the clarity of this part. The section "Comparison with single ionization mode models" was rewritten to make it clearer and less confusing. In addition, we have updated Figures 3.1 and 3.2 (now 3.3. and 3.4) to violin plots with a direct comparison to the model trained on both ionization modes at the same time. Both changes combined contribute to a clarification of this section and the presented results.

In the discussion, authors do mention in the discussion the imbalance in the ionization modes represented in the training data yet do not provide any quantification (e.g., ratio of ESI+ to ESI- data?).

We have now added these numbers also to the relevant discussion section:

"Because more spectra were available in positive ionization mode (519,580) than in negative ionization mode (145,594) this resulted in sampling more positive ionization mode pairs compared to negative ionization mode pairs. "

2.5.3 Finally, what do authors mean by Tanimoto bins having a “higher loss” in one model compared to another?

See the changes mentioned under 2.5.1. We hope these changes help in understanding what we meant.

2.6 Part of “Uncertainty Evaluation” Figure 3B: Can authors clarify what is on the y-axis in Figure 3C? Presumably, it represents the true MSE and the percentage of spectra represents the percentage after removing the top X% spectra based on their predicted MSE.

Indeed, it is the true MSE and the percentage of spectra represents the percentage after removing the top X% spectra based on their predicted MSE. We have now updated the figure and improved the clarity of the figure description.

a) Embedding evaluator b) Error prediction

c) Removing spectra with high predicted error

Figure 4: Results Embedding Evaluator. The embedding evaluator predicts if MS2DeepScore can make reliable chemical similarity predictions for an input spectrum. This embedding evaluator is trained to predict the mean squared error (MSE) for a spectrum. a) Architecture for the embedding evaluator. b) The predicted MSE is plotted against the true MSE for each spectrum in the test set. The test set contains 32052 spectra. c) Different thresholds are used to remove spectra from the test set with high predicted MSE. This results in 5 subsets of spectra with respectively 20, 40, 60, 80 and 100 % of the test spectra retained. The plot shows the real MSE per bin for predicting chemical similarity for the selected subsets of the test set. This shows that the accuracy of MS2DeepScore improves if the Embedding Evaluator is used to remove spectra with high predicted MSE. The MSE is calculated by taking the average MSE between all spectra of two molecules, followed by taking the average per Tanimoto bin.

2.7 Lines 306-307, authors state that spectra that have little similarity yet low quality tend to have very similar embeddings. How often does this occur?

We do not have a clear analysis of the frequency that this occurred, it was something we noticed for an experimental dataset, which inspired us to develop the embedding evaluator to recognize such cases.

While Figure 3b shows a strong correlation between true and predicted MSE, there are still many spectra that do not show strong agreement between the 2. In fact, it seems that predicted MSE values are generally a bit lower than the true MSE (many more points are falling between the x~y line).

The accuracy estimation is indeed not always perfect, but there is a strong correlation, which can help filter out the most uncertain predictions. At the moment, we do not have clear ideas for further improving the accuracy of these predictions. We are open for suggestions!

Have authors explored whether there the differences in true and predicted MSE are associated with the size or class of metabolites representing the spectra? The ones that have a high MSE, are they true or false positives?

Not exactly, but we have explored the relationship between the predicted MSE and precursor m/z and number of peaks. This showed some clear trends, like higher predicted MSE for spectra with a low number of fragments and a higher predicted MSE for smaller metabolites. The results can be found in Supplementary Figure 6.1.

2.8 While the example use case shown by authors is very interesting, it does not provide clarity on the utility of the tool. As a user, how does one find examples such as cholic acid? How does one best explore the data? UMAP or molecular networks? While the tool itself is theoretically useful, the practicality on how to use the tool is lacking.

This is indeed very important to clarify clearly. We have added additional case studies with extra examples and have added a workflow overview figure, including the use case of MS2DeepScore as suggested in comment 2.1. Combined we believe this better conveys the message on how to use this tool for data exploration and prioritization. We have also clarified the benefits of UMAP over molecular networks more clearly.

“MS2DeepScore also enables alternative visualization methods that overcome some limitations of molecular networking. One limitation of molecular networking is that it does not depict relationships between separate clusters. A second challenge with molecular networking is choosing a hard cut-off for when connecting an edge. If this threshold is set too high, relevant connections are not depicted, while if this threshold is set too low a hairball of connections can form, which makes it challenging to interpret the data. An alternative visualization method is using UMAP to visualize MS2DeepScore embeddings, this overcomes both these limitations of molecular networking.”

2.9 In the methods it is mentioned the improvements to MS2DeepScore 2.0 has “resulted in a substantial speed-up in the training of models”. There is no comparison metrics provided to show the improved speed.

When testing our most up-to-date model for the manuscript, we found out the speedup is not very substantial anymore. The model training using the old TensorFlow architecture took 19 hours for training both the positive ionisation mode and the negative ionisation mode mode, while the new cross-ionization mode model took 11.5 hours to train. When we wrote this line (in our initial preprint in 2024), we were still using a smaller model architecture (about 5 times smaller). The architecture of the model used now in our manuscript has about 105 million weights, while the original MS2DeepScore model had only about 5.3 million weights. Even though the training of models of the same size is now substantially faster, we do not think we can claim substantial speedup anymore, since moving to larger models cancels most of this effect. Therefore, we decided to remove the statement about substantial speed improvements. Thank you for noting and sorry for our oversight.

2.10 Supplementary Figure 1.1

2.10.1 What is the rationale for training on positive ionization mode data but using both ionization modes from the validation set?

The models tested here were trained on both ionization mode spectra. We made a mistake in the caption, this has now been corrected, sorry for the mistake.

a) All validation spectra

b) Validation set with > 4 peaks (intensity > 2%)

Supplementary Figure 1.1: Average RMSE per bin for model trained with different filtering of training spectra. For one of the models the training spectra only contained spectra that have at least 5 peaks at intensity >2%. The other model was trained on all spectra. The model trained on all spectra is used in the rest of the paper, since it resulted in best overall performance. Both models were trained with identical settings. a) Both models were validated with the standard validation set, containing all spectra. b) Both models were validated with the validation set only containing spectra with at least 5 peaks at intensity >2%.

2.10.2 The legend for this figure is unclear. For example (a) is referring to 2 models (authors write “both models”), but it is unclear which 2 models they are referring to.

We have updated the figure description to clarify this further.

2.11 Supplementary Figure 1.2 and 1.3: Can authors clarify what ‘dense layer dimensions’ mean? Additionally, why are they separate figures? The legend is exactly the same?

Thank you for checking the manuscript so carefully. This is a mistake that we unfortunately missed. The Figures are indeed very similar; they both test different deep learning model architectures. Supplementary Figure 1.2 changes the depth (the number of layers) of the deep learning model. Figure 1.3 uses a model that has a constant number of layers (only one layer), while varying the size of this layer. This was not described correctly in the Figure caption. In addition, the description was very limited. We have updated the Figure captions to correct the mistake, but also to give more explanation on what we exactly mean with the different model architectures. In addition to Figures 1.2 and 1.3, the other Figure captions in Supplementary 1 have also been changed to improve clarity.

The Figure legends now read like:

“Supplementary Figure 1.2: Models trained with varying model architectures. The model architecture was varied to select a suitable model architecture. The models all used the same hyperparameters as the final model, except for the number of layers and dimensions of these layers. The input layer always consisted of 10000 bins with a width of 0.1 Da and a final embedding size of 500. The legend shows the dense layers used between the input layer and the predicted embedding, e.g., 2000, 2000, 2000 is a dense neural network with 3 fully connected layers, each with 2000 nodes. The validation MSE was calculated over the spectra in the validation test set. To calculate the validation MSE, the pairs were binned in 10 equal Tanimoto score bins between 0 and 1, the MSE was calculated per bin and the average was taken over the 10 bins.”

“Supplementary Figure 1.3: Models trained with a single dense neural network layer varying the dense layer dimension. The model architecture was varied to select a suitable model architecture. The models all used a single dense layer between the input data and the predicted embedding. The dimension of this single layer was varied. The validation MSE was calculated over the spectra in the validation test set. To calculate the validation MSE, the pairs were binned in 10 equal Tanimoto score bins between 0 and 1, the MSE was calculated per bin and the average was taken over the 10 bins. “

Minor comments:

2.12 Lines 251-258: authors describe the inequality in the number of pairs evaluated per Tanimoto bin. The total number of pairs within a Tanimoto bin are not shown however in Figure 1 or other figures. Those are important to show and better interpret the density metrics.

We have replaced Figure 1 (Now Figure 2) with violin plots and bar plots showing the number of pairs.

The imbalance we discuss in 251-258, regards an imbalance in sampling during training. But it is true that during evaluation, there is also a kind of imbalance, since higher similarity pairs are a lot less frequent than low similarity pairs.

2.13 While supplementary figures are referenced by section (Supp Fig. 1, Supp Fig. 2, etc.), many are not referenced individually (Supp. Fig. 1.1, etc.) and are associated with distinct results and interpretations.

Thank you for noticing. We are now referencing almost all Figures individually in the main text. The only exceptions are the figures in section 2 and section 5. These are not referenced individually, since the interpretation of the results requires the combination of the entire section, including the additional text in this section.

2.14 Figure 1 and 3B: Consider having the same scale for both x- and y-axes, making it easier to see the highest density along the $y=x$ line.

Thank you for the suggestion. In Figure 3B we have made both axes equally, this indeed helps in seeing the correlation. Figure 2 was changed to the violin plots, and now also uses the same axis between 0 and 1. For Figure 2 this was achieved by setting negative predictions smaller than 0 to 0.

2.15 Figure 3B: please clarify in the legend how many spectra in total are evaluated. Consider changing all figures with densities (e.g., Fig. 1 and supp figures) to number of spectra or showing densities in main figures and number of spectra in supplementary figures.

The entire test set was used, which contains 32052 spectra. This was added to the figure caption. In addition all heatmaps were replaced with violin plots with bar plots showing the number of spectra.

2.16 There are a few inconsistencies with labeling of figures:

- o Line 115: Figure 4C should be Figure 1c
- o Line 360: Figure 1 should be Figure 4
- o Line 526: Supplemental Table 7.1 should be Supplemental Table 7.2

Thank you for noticing and apologies for these mistakes, they have been corrected.

2.17 Supp Materials, section “Balanced molecule sampling”. In first paragraph, authors refer to Figure 1A yet Figure 1A does not directly show the inequality of sampling.

This should have been Supplementary Figure 2.1a. We have corrected this. Thank you for noticing.

2.18 Legend of Figure 2: it would be useful to clarify explicitly that the MS2DeepScore scores are used as edges.

Good suggestion. We have added this to the description.

2.19 Lines 462-464: It seems that each embedding evaluated for the training of the model was that of the training data only? The wording is a bit unclear.

Yes, that is the correct interpretation. But the wording was indeed unclear. We have rephrased this section:

The “Embedding Evaluator” model is implemented using an Inception Time architecture⁴⁴ using Pytorch²⁵. The embedding evaluator is trained to predict the mean squared error (MSE). For each training spectrum, the MSE is calculated by randomly sampling 999 other training spectra and calculating the MSE over these 999 pair predictions. The model is trained to predict the MSE from the embedding of the spectrum. The conceptual idea here is that the Embedding Evaluator will learn to identify embeddings for low-quality or out-of-distribution input data. In later applications, the predicted embedding qualities can be used for uncertainty estimation.

2.20 ‘mzmine’ should be written as ‘MZmine’

We initially wrote it like that as well, but got corrected by the developers of mzmine, to spell it like mzmine. There are many different ways of spelling out there and you are correct that the paper about MZMine 3, spells it like that. But since the developers of mzmine changed official spelling for mzmine 4.0 to mzmine, we decided to stick with that here.

2.21 Supplementary Figure 5.3: Is this for the test set?

Yes, this was for the test set. It represented the same information as Figure 1, in a different way. Since we have now replaced Figure 1 with the violin plots, the boxplot does not add additional information, so it was removed from the supplementary information.

2.22 Supplementary Figure 5.1 and 5.2: What is the ‘n’ for each boxplot? Violin plot might show distributions a bit better?

Thank you for the suggestion, that reflects the data better indeed.

In addition, your suggestion on using a violin plot here inspired us to replace the heatmaps in Figure 1 with violin plots. We think this resolves many of the raised issues with the heatmap. So thank you for that suggestion!

Supplementary Figure 7.1: MSE per adduct type. For the test set the MSE per spectrum was calculated. Here the distribution of MSE per adduct type is given. Any adduct with less than 100 spectra is

Supplementary Figure 7.2: MSE per chemical class. For the test set the MSE per spectrum was calculated. Here the distribution of MSE per chemical class is given. The chemical class is determined by using ClassyFire¹, both molecules for which no ClassyFire annotation was available and chemical classes with less than 300 spectra are combined in the boxplot Unknown/other.

2.23 Supplementary Figures 6.1 and 6.2: title of the figure needs to be changed to say subset of test set.

Thank you, this has been changed.

2.24 'Uncertainty evaluation'

2.24.1 The authors have three panels in Figure 3 but only discuss Figure 3c. We have added references to all subfigures in the result section: "This model predicts the mean squared error (MSE) from the embedding of a spectrum, see Figure 4a. Figure 4b, shows that there is a strong correlation between predicted MSE and real MSE. Figure 4c shows the effect of removing the spectra with the highest predicted MSE. By filtering out spectra that have a high predicted MSE, the prediction accuracy between the remaining spectra increases."

2.24.2 Can the authors clarify when the authors say, 'We can improve prediction reliability by filtering out spectra with a high predicted MSE'. Does this mean that for each molecule pair, spectra with a high MSE are removed? Individual spectra are removed, not pairs of spectra. We have rephrased this to improve clarity. "Figure 4c shows the effect of removing the spectra with the highest predicted MSE. By filtering out spectra that have a high predicted MSE, the prediction accuracy between the remaining spectra increases."

2.24.3 It's unclear what the cleaning process for different analyses: for example, In Supplementary Figure 4.2a, the lowest number of fragments in the boxplot is presented as 1 but in the methods section under "Case Studies" it is mentioned that only MS2 spectra with more than four fragments are kept. Are these different data?

Yes, they are different data, in the specific case of Supplementary Figure 4.2a (now 6.2), these are artificially generated spectra. We do this to get some insights into what kind of information the embedding evaluator uses to predict the MSE.

However in other Figures like Supplementary Figure 6.1a we do use the test set, which is a subset of the annotated public data, which was not used for training. This test set was not filtered on a minimum number of fragments, while the Urine case study was. Initially we trained and tested the model with similar filtering of a minimum of 5 peaks. But after experiments shown in Figure 1.1, we found out that including all spectra during training (even if they are poorly fragmented) improved performance. For the newly added case studies, we did not do any filtering on the minimum number of peaks.

Reviewer #3 (Remarks to the Author):

REVIEWER COMMENTS

Reviewer #1 (Remarks to the Author):

1.2: The authors reply to my previous point, that "we do not think that the different method of chromatography [...] represent invalid examples. The advantage of using the fractions is that the metabolites had been partly separated and, in comparison to the raw urine, they can be more easily detected and therefore fragmented thanks to the reduced ion suppression."

And this is precisely the problem. If you use data generated to contain richer compound information ("easily detected", meaning that more compounds and with better MS/MS data will be recorded), you will likely find more examples of the utility of the tool. But if this is not the kind of data typically used in metabolomics, then the demonstration of the utility of the tool is biased, and might suggest that the utility of the tool is limited. I appreciate, however, that the authors have included other datasets.

We agree that it is important to showcase how our model can be applied to metabolomics studies that are representative for real use cases. In the previous round of revisions we added a case study on *Rumex sanguineus*, the data for this case study was collected with a conventional approach. The results obtained for this dataset are visualized in Figure 3b showing examples of similar compounds clustered closely together, also across ionization modes. The results are also available as an interactive html file, to allow for exploring the complete case study dataset and UMAP visualization. The results are comparable to the case studies performed on the fractionated data. We hope that showing the results on this plant case study gives a good overview of how our dual ion mode model is expected to perform on conventional datasets.

We would also like to stress that the fractionation - in principle - has no or only little influence on the quality of the MS/MS spectra itself, it will mainly increase the number of ions for which tandem mass spectra are recorded in the sample. These additional mass spectra might not have been recorded in standard metabolomics experiments, since they would not be selected as the most abundant MS1 peak. These additional mass spectra that are normally suppressed likely belong to compounds that are not easily detected and are therefore less likely to appear in the public datasets we used for training. Therefore, this makes predictions for these mass spectra more challenging for our model, instead of easier.

1.3. My point was specifically focused on comparing single ionization mode vs the cross/dual mode presented in the study. From the response, the authors understood my concern, but instead argue that this is not what they were trying to demonstrate: they wanted to show the improvement over the previously published version. From the figure, the performance improvement is subtle. In any case, I disagree with this response. The tool's ultimate goal is to make use of the potentially richer information that can be extracted when jointly using pos and neg modes, compared to single (either pos or neg) mode. And the authors, with their own results, showed that there is virtually no difference. Therefore, the utility of the method is questionable, considering (see previous point) that the authors even used data intended to improve this scenario.

We appreciate the reviewer's clarification of their earlier point and we understand that our earlier description may have given the impression that our method would require paired positive and negative ionization mode spectra from the same compound. This is not the case. Instead, the model predicts chemical similarity between ANY two spectra, including a positive-negative pair originating from different compounds.

In the previous round of revisions we have therefore made several changes to improve the clarity of explaining how the method works, most importantly by adding Figure 1 as an overview figure to visually show what MS2DeepScore 2 does.

Regarding the reviewer's concern about the comparison between single-mode and dual-mode training: the purpose of Supplementary Figures 4.3, 4.2c, and 4.1a is simply to confirm that incorporating both ionization modes during training does not substantially degrade performance on the conventional within-mode chemical-similarity task. These analyses do not involve dual-mode prediction and were included only for completeness.

The central contribution of the method is the newly introduced cross-ionization-mode similarity prediction. Supplementary Figures 4.1b and 4.2b show that models trained on a single ionization mode perform, as expected, very poorly on this task, whereas the dual-mode model performs substantially better and recovers meaningful cross-mode chemical relationships. This demonstrates that the method provides genuinely new functionality rather than incremental improvement.

The cross ionization modes have substantial practical impact by enabling sample visualizations that combine both positive and negative ionization mode spectra in one overview. All previous methods have to create two separate overviews for each ion mode, since no meaningful predictions can be made between a positive and negative ionization mode spectrum. We show this utility by showing cross ion mode molecular networks and UMAP visualizations, where manual validation confirms that the new cross-ionization mode connections are chemically meaningful.

We acknowledge that there are still limitations to this new functionality. In particular, for high similarity across ionization modes the model predicts slightly lower similarity scores than for within-mode pairs. As a consequence, some highly similar cross-mode connections may not be clustered together when using a fixed threshold. In the response on the next comment we go into more detail and have improved the discussion to more clearly highlight the current limitations of the method. Still, already in its current form, the method enables cross-ion mode networking and visualization that were previously impossible, we believe this clearly demonstrates the newly added utility.

1.4 While I appreciate the addition of Fig 10.2, for the case of Fig 10.1, which is included in response to my concern, the figure shows the same example multiple times, therefore, not addressing my concern.

We understand that it may seem like some spectra are duplicated in Figure 10.1, since some of the molecules are duplicated and some spectra are almost identical, but they all were unique spectrum pairs. Many molecule pairs were indeed repeated, because multiple MS² spectra were generated for the same molecule. We acknowledge that this did not adequately address the raised concern, and we now replaced figure 10.1 with a figure where we only show one of the spectrum pairs per unique molecule pair.

Supplementary Figure 10.1: Spectrum comparisons of positive and negative ionization mode spectra for which MS2DeepScore predicts high chemical similarity. The spectra are from the urine case study. Positive ionization mode spectra are blue and negative ionization mode spectra are orange.

Now, the authors include a new Fig 2 where they show a "similarity correlation" plot to demonstrate the tool's performance. The standard deviation is very high and this analysis only shows a general performance trend but does not analyze the performance in depth. I suggest including a qualitative analysis where top-k results are shown, e.g., in how many cases the most similar molecule is predicted within top-1, top-3, top-N predictions; and also include an additional quantitative analysis to show the relative true-predicted similarity error. These analyses should be triplicated to compare pos vs neg vs the dual mode to show the advantages of this dual mode. I am referring to the actual application of the tool, not whether the tool has been trained with only one mode or dual mode data.

Thank you for the suggestions on how to further improve on communicating the utility of the model. We have added additional analysis and more extensive discussion on the practical utility of MS2DeepScore.

We realize that what was mostly lacking was a clear discussion and distinction between the different types of downstream use cases for using this new cross ionization mode model. One is organizing and visualizing samples with for instance molecular networking or UMAP visualization and the second is for annotation by searching for exact matches or analogues in a reference library with annotated spectra. The currently strongest direct utility of our model lies in the organization and visualization of samples across ionization modes. For cross ionization mode analogue searching, our model is a key enabler to do this in the future, but it is currently not directly practically useful for an analogue search. We did observe the same limited utility for analogue searching in the original MS2DeepScore paper as well. That said, we have run the suggested analysis. This shows that MS2DeepScore does add new cross ion mode scores when utilized in a setting similar to molecular networking, however when applying it in a naïve analogue search (top 1 highest score) the addition of cross ion mode predictions is almost negligible. In a different project, we are planning to implement the new model into MS2Query to finetune and optimize its use for analogue search. We have added new analysis to the supplementary information and we added the following section to reflect the above considerations and findings, and discuss current limitations and future recommendations:

“Spectral similarity scores like MS2DeepScore are used for various downstream applications; however, we recognize two main ones: 1) organizing and visualizing samples by identifying high chemical similarity within a dataset, and 2) library searching for exact matches or structurally related analogues. The here introduced cross-ionization-mode prediction functionality primarily benefits the first task. Our case studies show that MS2DeepScore correctly predicts multiple high similarities between positive and negative ionization mode spectra. These edges correctly bridge clusters that were previously disjoint when using only within-mode predictions.

However, an important limitation remains: the number of cross-ionization-mode edges that exceed a similarity score of 0.85 and fall within the top-10 highest scores is low. Lowering the threshold for the minimum predicted similarity score increases the number of cross-ionization-mode connections, but this is not generally recommended, since this also substantially increases the number of false positives. This then leads to the molecular network becoming a hairball, hindering effective data analysis. These limitations are less consequential in the UMAP-based visualization, where no strict threshold is required. Here spectra from positive and negative ionization mode spectra are integrated smoothly into a unified chemical space, without the need for specific thresholds.

The new cross-ionization mode capability is the first step to enable cross ionization mode library searching for analogues in the future. However, when implemented in a naïve approach of selecting the top 1 highest candidate, there is no added benefit from using a cross ionization mode model (See supplementary Section 11). There is only a small number of cases where the predicted score is both above the threshold of 0.85 and higher than all the within ion mode connections. Our previous work for MS2Query showed that the accuracy of analogue searching using chemical similarity prediction tools like MS2DeepScore can be improved substantially by reranking the best candidates based on the similarity predictions for multiple closely related library molecules. We therefore recommend an approach like introduced in MS2Query as a starting point to enable reliable cross ionization mode analogue searching. At present, the principal value of cross-ion-mode MS2DeepScore predictions lies in within-sample organization and visualization across ionization modes, where the method already provides clear and actionable benefits.”

To the supplementary information, we added the following section:

Supplementary Section 11: Benchmarking specific use cases

Additional benchmarking tests are performed to illustrate accuracy on downstream tasks like molecular networking and analogue searching. To this end, the test set was used as described in the methods section, “Input data filtering and splitting”. This is a random subset of the annotated public libraries. For each query spectrum, the best possible analogue search hit is selected from the rest of the test set, including both the positive and negative spectra. This selection is based on the highest Tanimoto score, excluding any exact matches. After selecting the best possible analogue, the MS2DeepScore model is used to rank all mass spectra in the test set for each query spectrum. The percentage that the best possible analogue is ranked in the top 1, top 3 or top 10 is given. The analysis is repeated for searching in each ion mode or searching in both ionization modes. The result is weighted to ensure each unique InChIKey counts equally. Supplementary table 11.1 shows that searching in both ionization modes does

not impact how often the best hit is found in the top 1, but results in an increased frequency of selecting the best possible match within the top 10.

Query ion mode	Library ion mode	Best analogue in top-1	Best analogue in top-3	Best analogue in top-10
Negative	Negative	1.1%	4.0%	8.4%
Negative	Negative + positive	1.1%	3.9%	9.6%
Negative	Positive	2.0%	3.4%	6.0%
Positive	Positive	3.9%	8.0%	12.9%
Positive	Positive + negative	3.9%	8.3%	13.4%
Positive	Negative	1.2%	1.8%	3.2%

Supplementary Table 11.1: Percentage of best possible analogue in top k. The best possible analogue is selected from all available InChIKeys, in both positive and negative ionization modes.

To qualitatively illustrate the accuracy of the top predictions, additional benchmarking is performed. From the test set, a single spectrum is selected at random for each unique molecule. From this subset, 924 spectra are sampled from both the positive and negative ion mode spectra to have an equal number of mass spectra for each ion mode. Predictions are made between all the selected mass spectra using the new MS2DeepScore model, both within and across ionization modes. The predictions are ranked, and the top k highest predictions are selected that are above a minimum threshold. To illustrate a molecular networking scenario, we use the top 10 hits, and to illustrate an analogue searching scenario, we use the top 1 hit. For these highest-ranked predictions, the error is computed by subtracting the real Tanimoto score from the predicted Tanimoto score. From all 1848 mass spectra the top k are selected that have a score higher than the minimum threshold. The top predictions are separated into four cases, pos-pos, neg-neg, pos-neg, and neg-pos.

Supplementary Figure 11.1 illustrates a case like analogue searching where the top 10 hits are selected, that have a minimum threshold of 0.85. These are the same settings used as in the case studies illustrated in Figure 3. The test set used here is a more challenging case than the case studies, since the molecules in this test set are randomly selected from all available training data and are therefore expected to be much more diverse than a normal sample, therefore having less available high chemical similarity pairs within the dataset than a normal dataset. Therefore, the case studies represented

in Figure 3 give a more realistic overview on how the tool is expected to perform on real data to create a molecular network.

In Supplementary Figure 11.1 and 11.2 we use a minimum threshold of 0.85. Using such a threshold is recommended, since not using a threshold results in too many false positives. This is illustrated in Supplementary Figure 11.3 and 11.4, here no threshold is used, which results in a lot of false positives.

Supplementary Figure 11.1: Error for top-10 highest predictions above 0.85. For each mass spectrum, the top 10 highest predictions are selected that have a predicted score of at least 0.85. The error is the difference between the predicted score and the real Tanimoto score.

Supplementary Figure 11.2: Error for top-1 highest predictions above 0.85. For each mass spectrum, the top 1 highest predictions are selected that have a score of at least 0.85. The error is the difference between the predicted score and the real Tanimoto score.

Supplementary Figure 11.3: Error for top-10 highest predictions, without filtering on a minimal score. For each mass spectrum, the top 10 highest predictions are selected, without filtering on a minimum score. This is not recommended, since it results in many false positives. It is recommended to set a high minimal chemical similarity score. The error is the difference between the predicted score and the real Tanimoto score.

Supplementary Figure 11.4: Error for top-1 highest predictions, without filtering on a minimal score. For each mass spectrum, the top 1 highest predictions are selected, without filtering on a minimum score. This is not recommended, since it results in many false positives. It is recommended to set a high minimal chemical similarity score. The error is the difference between the predicted score and the real Tanimoto score.

Overall, my concerns still stand, and they are aligned or very similar to those of Reviewer #2, a fact that reinforces their validity. One of the other reviewers' concerns is about the shared molecules across modes. The authors' response included an analysis of a database instead of using real data samples. This also shows that the authors can not demonstrate the advantage of combining both modes. Taking the results, my points and the authors' response together, it seems that the tool is limited by the poor information that the suggested dual mode can contribute to the identification of molecules. While I appreciate the study within this manuscript, I can not endorse this manuscript for publication in Nat. Comms for the same reasons stated in my previous report.

We are grateful for the reviewer's careful feedback, which has helped us to clarify both how MS2DeepScore works and what its current, direct utility is. In the revised manuscript we now more clearly distinguish between (i) organizing and visualizing samples across ionization modes and (ii) using similarity scores for analogue or identification workflows, and we explicitly discuss the current limitations for the latter and directions for future work. We believe that, with these clarifications and the additional analyses, the manuscript now more transparently demonstrates the concrete benefits of the dual-mode model for studies that acquire both positive and negative ionization mode data. In its current form, the work is positioned as a method that enables integrated cross-mode sample organization and visualization, and as a necessary first step toward more advanced cross-mode analogue search methods.

Reviewer #2 (Remarks to the Author):

The authors have greatly improved the clarity of the results, methods and manuscript in general. Please ensure that the answers to the queries provided are included in the results and discussion as appropriate. Several minor specific points to note:

Thank you for the clear feedback, we agree this has substantially improved the strength and clarity of the manuscript. We have now made sure that all answers are incorporated well in the main text.

- Regarding point 2.2.1: the replacement of the heatmaps by the violin plots is very helpful. However, because the scales are different (necessarily so to create the violins), it's difficult to visualize where the $y=x$ line falls. Can this be somehow displayed for easier reading of the plots?

Thank you for the suggestion. We have added a diagonal line showing the $y=x$ line to the revised Figure 2.

Figure 2: Dual-ionization mode MS2DeepScore model predicts chemical similarity between and across ionization modes. A test set of 32052 spectra is used, which were not used to train the model. Predictions are made between all test spectra, followed by taking the average per unique molecule pair. The violin plots show the kernel density estimation (KDE) of the predicted values, the black lines represent the median and the 1st and 99th percentile for each bin. The dashed line indicates the $y = x$ line. The bar plot on the top shows the log-scaled count of the number of unique molecule pairs in each bin with the corresponding chemical similarity. The metric used for chemical similarity prediction is the Tanimoto score between Daylight fingerprints. a) Predictions between pairs of positive ionization mode spectra. b) Predictions between pairs of positive and negative ionization mode spectra. c) Predictions between pairs of negative ionization mode spectra.

- Regarding point 2.2.3: while authors reference Supp Figures 7.1 and 7.2 in the text and provide an explanation in the rebuttal, they do not explain in the text that certain adducts have higher MSE. It would be useful to clarify this in the text. The same comment holds for Supp Figure 6.1.

We agree that it is helpful to the reader to mention more details about these results from the supplementary information. We have now extended the description in the results section:

“In addition to this benchmarking we analyzed the performance of MS2DeepScore for different adducts and different compound classes, these results can be found in Supplementary Figures 7.1 and 7.2. These analyses give insights into the general patterns of the strengths and weaknesses of the MS2DeepScore model. “Lipids and lipid-like molecules” and “nucleosides, nucleotides and analogues” had slightly better performance, while organic oxygen compounds had a higher error. Supplementary Figure 7.2 shows that for the adducts $[M+Na]^+$ and $[M+K]^+$ the accuracy was lower than average.”

“Additional analysis in Supplementary Figures 6.1 and 6.2 tested the correlation between predicted MSE and features like the number of fragments, precursor m/z , ionization mode, and signal intensities. This showed some clear trends, like a higher predicted MSE for spectra with a low number of fragments and a higher predicted MSE for smaller metabolites.”

- Regarding 2.4.1: Authors clarify in the rebuttal that users need to manually differentiate between exact matches and closely related molecules and could use RT and mass differences to help with this task. This could be clarified in the text, as well as potential future automation of this.

Thank you for the suggestion, we agree that it is valuable to discuss this in the manuscript. We have added the following paragraph to the Discussion section:

“Other methods like Ion Identity Molecular Networking²⁰ and MolNotator¹⁹ allow users to combine mass spectra of different adducts that belong to the same metabolite by using mass difference and retention time alignment. MolNotator applied this principle to merge molecules across ionization modes. A limitation of these methods is that only metabolites can be linked across ionization modes if two mass spectra were recorded for the same metabolite in both the positive ionization mode and negative ionization mode. Since MS2DeepScore can also predict chemical similarity across ionization modes if two metabolites are not identical, additional connections can be made between highly similar molecules across ionization modes. However, a limitation of MS2DeepScore is that no clear distinction is made between identical metabolites and closely related metabolites across ionization modes. Future work could combine approaches like Ion Identity Molecular Networking with MS2DeepScore to connect and confirm identical matches across ionization modes and to clearly distinguish closely related molecules from identical molecules.”